# Spatiotemporal organization of branched microtubule networks

**Akanksha Thawani[1], Howard A Stone[2]\*, Joshua W Shaevitz[3,4]\*, Sabine Petry[5]\***

[1]Department of Chemical and Biological Engineering, Princeton University, Princeton, United States; [2]Department of Mechanical and Aerospace Engineering, Princeton University, Princeton, United States; [3]Lewis-Sigler Institute for Integrative Genomics, Princeton University, Princeton, United States; [4]Department of Physics, Princeton University, Princeton, United States; [5]Department of Molecular Biology, Princeton University, Princeton, United States

**Abstract** To understand how chromosomes are segregated, it is necessary to explain the precise spatiotemporal organization of microtubules (MTs) in the mitotic spindle. We use *Xenopus* egg extracts to study the nucleation and dynamics of MTs in branched networks, a process that is critical for spindle assembly. Surprisingly, new branched MTs preferentially originate near the minus-ends of pre-existing MTs. A sequential reaction model, consisting of deposition of nucleation sites on an existing MT, followed by rate-limiting nucleation of branches, reproduces the measured spatial profile of nucleation, the distribution of MT plus-ends and tubulin intensity. By regulating the availability of the branching effectors TPX2, augmin and γ-TuRC, combined with single-molecule observations, we show that first TPX2 is deposited on pre-existing MTs, followed by binding of augmin/γ-TuRC to result in the nucleation of branched MTs. In sum, regulating the localization and kinetics of nucleation effectors governs the architecture of branched MT networks.

DOI: https://doi.org/10.7554/eLife.43890.001

**\*For correspondence:**
hastone@princeton.edu (HAS);
shaevitz@princeton.edu (JWS);
spetry@Princeton.EDU (SP)

**Competing interests:** The authors declare that no competing interests exist.

## Introduction

The mitotic spindle is a complex and dynamic macromolecular machine that is responsible for segregating chromosomes during cell division (*McIntosh et al., 2012*; *Prosser and Pelletier, 2017*). It is currently thought that the microtubule (MT) architecture in the spindle is governed by the spatiotemporal profile of MT nucleation and the action of molecular motors (*Brugués et al., 2012*; *Needleman et al., 2010*; *Oriola et al., 2018*). Spindle MTs originate from spindle poles, chromosomes, and from the lateral surface of pre-existing spindle MTs in a process termed branching MT nucleation (*Dinarina et al., 2009*; *Duncan and Wakefield, 2011*; *Hayward et al., 2014*; *Heald et al., 1996*; *Maresca et al., 2009*; *Meunier and Vernos, 2016*; *Petry et al., 2013*; *Piehl et al., 2004*; *Prosser and Pelletier, 2017*; *Tulu et al., 2006*). Because of the high density of MTs resulting from these nucleation pathways, resolving MT nucleation events in the spindle is challenging (*Brugués et al., 2012*; *Burbank et al., 2006*; *Decker et al., 2018*; *Kaye et al., 2017*; *Needleman et al., 2010*; *Oh et al., 2016*; *Prosser and Pelletier, 2017*; *Redemann et al., 2017*; *Tulu et al., 2006*; *Yang et al., 2007*).

Common to all MT nucleation pathways is the universal nucleator, gamma-tubulin ring complex (γ-TuRC) (*Kollman et al., 2011*; *Tovey and Conduit, 2018*), together with XMAP215/chTOG/Stu2 (*Flor-Parra et al., 2018*; *Gunzelmann et al., 2018*; *Thawani et al., 2018*). γ-TuRC has been proposed to be positioned and turned on at specific sites and times respectively, for each pathway during spindle assembly (*Kollman et al., 2011*) and a number of factors that perform these functions have been identified (*Alfaro-Aco et al., 2017*; *Choi et al., 2010*; *Cota et al., 2017*; *Kollman et al., 2010*; *Liu et al., 2014*; *Tovey and Conduit, 2018*). Yet, how the localization or activation of the

nucleators results in the spatiotemporal arrangement of MTs in the mitotic spindle remains to be demonstrated.

We sought to explore the nucleation profile and the role of localization and activation of nucleators in branching MT nucleation. Branching MT nucleation is an important spindle assembly pathway that nucleates new daughter MTs from the surface of pre-existing ones in a narrow range of angles (*Kamasaki et al., 2013*; *Petry et al., 2013*) and, in isolation, creates dense, wedge-shaped networks with uniform polarity (*Petry et al., 2013*). Loss of branching components causes reduced MT mass within the spindle body (*Goshima et al., 2008*; *Kamasaki et al., 2013*; *Petry et al., 2011*), instability of kinetochore fibers (*Bucciarelli et al., 2009*; *Lawo et al., 2009*; *Uehara et al., 2009*), insufficient tension during chromosome segregation (*Uehara et al., 2009*), and chromosome misalignment (*Goshima et al., 2008*). Most importantly, branching is the major nucleation pathway in spindles that form either without centrosomes (*Petry et al., 2011*) or in larger spindles (e.g. in *Xenopus* egg extracts), where it generates most of the MT mass (*Decker et al., 2018*). Nucleation of branched MTs requires several key molecules. The multisubunit complex augmin has been characterized as a targeting factor that binds to the nucleator γ-TuRC and recruits it to existing MTs in several model systems (*Kamasaki et al., 2013*; *Nakaoka et al., 2012*; *Petry et al., 2011*; *Sánchez-Huertas et al., 2016*; *Song et al., 2018*). The protein TPX2 is necessary for branching MT nucleation (*Petry et al., 2013*) and was recently proposed to activate γ-TuRC (*Alfaro-Aco et al., 2017*). Although the necessary molecules have been identified, it remains unclear how they establish a spatial pattern of branching that results in the characteristic tree-shaped networks, particularly when these proteins are thought to be homogeneously distributed in solution or along MTs (*Alfaro-Aco et al., 2017*; *Petry et al., 2013*; *Song et al., 2018*). Most importantly, the exact hierarchy and role of TPX2 and augmin in the branching pathway remains to be determined. Uncovering the building plan for branched MT networks will not only provide insights into the molecular mechanism of this pathway, but also identify how MT nucleation is regulated to construct the mitotic spindle.

In this study, we use *Xenopus laevis* egg extracts to study branched MT networks at single MT resolution. By measuring the spatiotemporal profile of branching MT nucleation and the shape of mature networks, we demonstrate a bias in the nucleation profile from older MT regions near the minus-ends, which is consistent with a two-step, sequential kinetic model involving rate-limiting nucleation from deposited nucleation sites. We establish that first TPX2 participates in depositing nucleation sites on individual MTs, followed by augmin/γ-TuRC binding to nucleate branched MTs, which explains the architecture of branched MT networks.

## Results

### High-resolution analysis of branched microtubule networks

We used total internal reflection fluorescence microscopy to study the evolution of branched MT networks in *X. laevis* egg extracts (*Petry et al., 2013*). Addition of the constitutively-active small GTPase RanQ69L to egg extract induces the nucleation of new MT branches from pre-existing MTs. To determine the effect of nucleation on network architecture without the complication of active MT transport by molecular motors, we further added the ATPase inhibitor vanadate. Nucleation events and MT growth were resolved by time-lapse imaging of fluorescently-labeled tubulin and end-binding protein 1 (EB1), which labels the growing MT plus-ends (*Figure 1A*; *Video 1*). Upon careful analysis, we consistently observed that branching events occurred primarily near the minus-ends of pre-existing MTs and were excluded from the vicinity of plus-ends (*Figure 1A*, *Video 1*).

We developed an image analysis procedure for automated tracking and reconstruction of branched MT networks (*Figure 1—figure supplement 1*; *Video 2*). We first tracked the position of EB1-mCherry-labeled MT plus-ends (*Applegate et al., 2011*; *Jaqaman et al., 2008*). While this is similar to previous tracking to obtain MT growth parameters (*Gutiérrez-Caballero et al., 2015*; *Henty-Ridilla et al., 2017*; *Ishihara et al., 2016*), it is insufficient to deduce the complete MT trajectory because EB1 spots disappear when the plus-end undergoes catastrophe or pause (*Figure 1—figure supplement 1B*, left). To obtain complete MT tracks (*Figure 1—figure supplement 1B*, right), we merged gaps in EB1 tracks based on the presence or absence of MT fluorescence in the gaps, as well as using known characteristics of plus-ends, such as their polymerization in a straight line at a specific speed (*Figure 1—figure supplement 1*; *Video 2*).

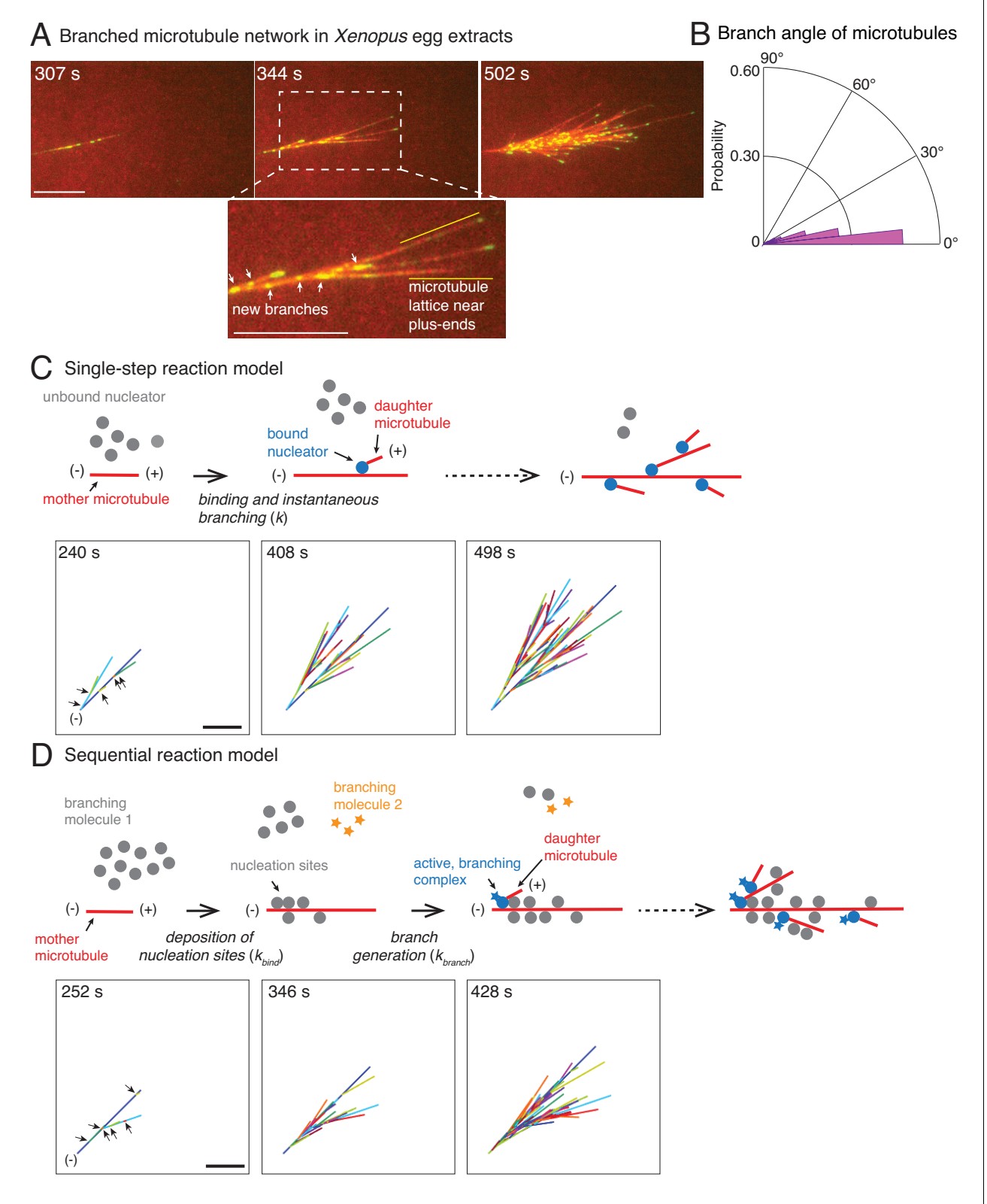

**Figure 1.** Branched microtubule networks and stochastic models for their assembly. (**A**) Branched microtubule (MT) networks were generated in *Xenopus* egg extracts with 10 µM RanQ69L, and time-lapse is displayed for one representative branched network. MTs were labeled with Cy5-tubulin (red) and their plus-ends with EB1-mCherry (pseudo-colored as green). 0 s represents estimated nucleation of the first mother MT. Scale bar, 10 µm. The highlighted region shows that new nucleation events (marked by EB1 spots) occur near the minus-ends and exclude the lattice near the growing

*Figure 1 continued on next page*

*Figure 1 continued*

plus-ends. The experiment was repeated with more than ten independent egg extract preparations. (B) Angle of branching for all branching nucleation events was calculated as described in Materials and methods. Polar histogram of n = 339 measurements from 19 branched networks is plotted. The median branch angle is 0° with a standard deviation of 9°. See Figure supplements and *Videos 1–2*. (C) Schematic representation of biochemical scheme for the single-step model. Free, inactive nucleators (grey) bind to existing MT lattice (red) and instantaneously nucleate a new daughter MT upon this binding (blue nucleators). Recursion of this process results in branched structures. Time-lapse of a representative stochastic simulation of the model is displayed with parameters provided in *Figure 1—figure supplement 3A*. Individual MTs are labelled in rotating color scheme. Arrows denote the positions of nucleation sites for first five branched MTs. Scale bar, 10 μm. See *Figure 1—figure supplement 3A* and *Video 3*. (D) Biochemical scheme for the sequential model. Molecules of branching effector 1 (grey) bind to existing MT lattice (red) and deposit nucleation sites. Subsequent binding of branching effector 2 (yellow stars) results in nucleation of daughter MTs. Time-lapse of a representative stochastic simulation of the model is displayed with parameters provided in *Figure 1—figure supplement 3A*. Individual MTs are labelled in rotating color scheme. Arrows denote the positions of nucleation sites for first six branched MTs. Scale bar, 10 μm. See *Figure 1—figure supplement 3A* and *Video 4*.

DOI: https://doi.org/10.7554/eLife.43890.002

The following figure supplements are available for figure 1:

**Figure supplement 1.** Hybrid tracking of branched microtubule networks.
DOI: https://doi.org/10.7554/eLife.43890.003
**Figure supplement 2.** Characterization of branched microtubule networks.
DOI: https://doi.org/10.7554/eLife.43890.004
**Figure supplement 3.** Stochastic models of branched microtubule networks.
DOI: https://doi.org/10.7554/eLife.43890.005

We used this data to quantify properties relating to MT nucleation and growth in branched networks (*Figure 1—figure supplement 2A*). The speed of plus-ends during growth was 8.3 ± 2.5 μm min$^{-1}$ (*Figure 1—figure supplement 2B*), similar to previous reports (*Tirnauer et al., 2004*). The net plus-end speed over the entire MT trajectory, including growth, catastrophe, and pause phases, was $v_{pe}$ = 5.3 ± 1.4 μm min$^{-1}$ (n = 115 MT tracks). The number of MTs in the network, counted via the number of EB1 comets where each EB1 comet marks a single MT, increased exponentially over time with an effective autocatalytic nucleation rate of 0.54 ± 0.06 min$^{-1}$ (n = 11 branched networks, *Figure 1—figure supplement 2C*). An important measure for the network architecture is the MT length distribution. MT lengths were exponentially distributed (*Figure 1—figure supplement 2D–E*) with an average length that approached a steady value over time (*Figure 1—figure supplement 2D*, red curve). Thus, branched MT networks are in a bounded growth regime, in agreement with previous measurements made in the spindle (*Dogterom and Leibler, 1993*; *Ishihara et al., 2016*; *Verde et al., 1992*). Lastly, we assigned each new daughter MT to its most probable mother MT and obtained the distribution of branching angles (*Figure 1B*). We find that MTs branch at a shallow angle with 0° median angle and a standard deviation (s.d.) of 9°. This is in agreement with previous observations (*Petry et al., 2013*) and similar to MT branching in neurons (*Sánchez-Huertas et al., 2016*). Altogether, these experimentally-derived parameters allowed us to generate a model for the formation and growth of branched MT networks.

## Stochastic model for branched microtubule networks

We start with the simplest stochastic model for autocatalytic nucleation of MTs. In this 'single-step model', a pool of MT nucleators bind to existing MTs and immediately generate a

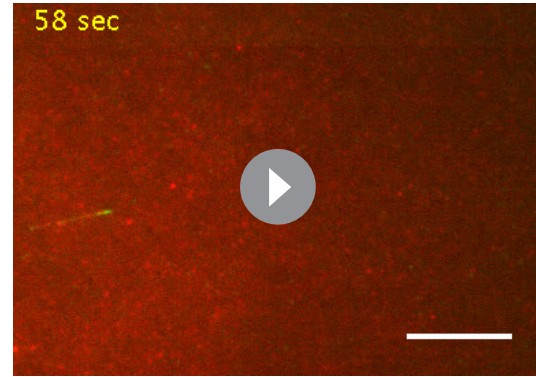

**Video 1.** Assembly of branched microtubule networks in *Xenopus* egg extracts. Branched microtubule (MT) networks were generated in *Xenopus* egg extracts with 10 μM RanQ69L, and time-lapse is displayed for one representative branched network. MTs are labeled with Cy5-tubulin (red) and their plus-ends with EB1-mCherry (pseudo-colored as green). Sodium orthovanadate was added to prevent motor-mediated MT gliding. Time-point 0 s represents the estimated nucleation time of first mother MT. Scale bar, 10 μm.
DOI: https://doi.org/10.7554/eLife.43890.006

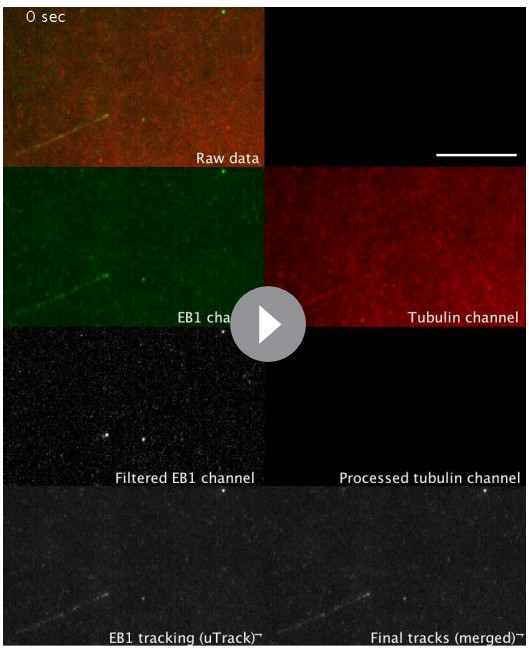

**Video 2.** Tracking of microtubules in branched networks. MTs in branched networks were tracked. Branched networks ('raw data') were imaged in *Xenopus* egg extracts with fluorescently-labeled plus-tip protein EB1-mCherry (green, displayed as 'EB1 channel') and Cy5-labeled tubulin ('tubulin channel'). EB1 comets were pre-processed using temporal median filter ('filtered EB1 channel') and tracked using comet detection and plus-tip tracking modules of *uTrack* ('EB1 tracking'). Tubulin intensity was pre-processed as described in *Figure 1—figure supplement 1* ('processed tubulin channel') and used to close the gaps in EB1 tracks using greedy optimization algorithm to obtain complete MT trajectories in the branched network ('final tracks'). Scale bar, 10 µm. The tracking procedure is described in *Figure 1—figure supplement 1* and Materials and methods section. MATLAB software for tracking is provided in *Supplementary file 1* .
DOI: https://doi.org/10.7554/eLife.43890.007

mean length of mother MTs at the first branching event measured experimentally (*Figure 1—figure supplement 3A*). The sequential reaction model also results in branched networks that resemble those observed in experiments (*Figure 1D*). Three realizations out of thousands of simulations are displayed in *Figure 1—figure supplement 3B* and *Video 4*.

## Measuring the first branching events in branched microtubule networks

To characterize the spatial organization of MTs in branched networks and distinguish between

daughter MT (*Figure 1C*). The binding of nucleators is assumed to occur with one effective rate-limiting reaction step ($k$). Simulations of this scheme use two measured parameters, the net plus-end growth speed ($v_{pe}$) and the branching angle distribution, and one fit parameter, the binding rate of the nucleators. A binding rate of nucleators to MTs, $k = 1.1 \times 10^{-3}$ molecules $\mu m^{-1}$ $sec^{-1}$, produced the mean length of mother MTs before the first branching event that matched with experiments (*Figure 1—figure supplement 3A*). The resulting network visually displayed the overall characteristics of the experimental branched networks (compare *Figure 1C* to *Figure 1A*). Thousands of simulations were conducted and three realizations are displayed in *Video 3* and *Figure 1—figure supplement 3B*.

A large body of research suggests that MT nucleation is highly regulated in the spindle through localization and activation factors of the MT nucleator (*Kollman et al., 2011*). Because multiple proteins necessary for branch formation have been identified and proposed to regulate MT nucleation (*Alfaro-Aco et al., 2017*; *Petry et al., 2013*; *Song et al., 2018*; *Thawani et al., 2018*), we also simulated a two-step model of branching. In this 'sequential reaction model', the first reaction step involves binding of one of the branching proteins to MTs which marks the nucleation sites, followed by a subsequent reaction step where a second branching protein generates the daughter MT (*Figure 1D*). Similar to the single-step model, we input the growth speed and the branching angle distribution, and fit two model parameters: the binding constant for the first step, $k_{bind} = 0.1$ molecules $\mu m^{-1}$ $sec^{-1}$, and the branching rate, $k_{branch} = 2.5 \times 10^{-4}$ $sec^{-1}$. The product of these two rate constants sets the effective nucleation rate in the system, which was fixed to match the

0 sec

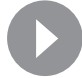

**Video 3.** Simulation of single-step reaction model. Three stochastic simulations of branched networks using the single-step model are displayed with parameters specified in *Figure 1—figure supplement 3A*. Individual MTs are labelled with a rotating color scheme. Scale bar, 10 µm.
DOI: https://doi.org/10.7554/eLife.43890.008

the single-step versus the sequential models, we measured the position of the first branching nucleation event on each mother MT (*Figure 2—figure supplement 1A*). We find that the first branching nucleation event is more likely to occur from older lattice regions near the minus-end of the mother MT (blue curve in *Figure 2A–B* and *Figure 2—figure supplement 1B*), supporting our previous visual observation (*Figure 1A*). Surprisingly, not a single nucleation event was observed in the region 2.1 μm from the plus-end of the mother MT (n = 381 measurements), resulting in a dead-zone for branching near the plus-end. Most branching events occurred 5–15 μm away from the mother's plus-end (*Figure 2A*, blue curve). We used a non-dimensional measure of this bias in branch location by dividing the distance between the branching site and the mother's minus-end by the total length of the mother MT at the time of branching. The probability of branching as a function of this rescaled distance decreases sharply from its highest value near the mother's minus-end to zero at the mother's plus-end (*Figure 2B*, blue curve).

**Video 4.** Simulation of the sequential reaction model. Three stochastic simulations of branched networks using sequential model are displayed with parameters specified in *Figure 1—figure supplement 3A*. Individual MTs are labelled with a rotating color scheme. Note that new branched MTs nucleate preferentially from older MT lattice near the network origin. Scale bar, 10 μm.
DOI: https://doi.org/10.7554/eLife.43890.009

We next compare these probability distributions using the two different computational models (*Figure 2A–B*). The single-step model does not display an exclusion zone near the plus-end (*Figure 2A*, red curve) or a non-dimensional bias (*Figure 2B*). In a single reaction step that includes nucleator binding and instantaneous branch formation, branches are equally likely to occur anywhere on the mother MT. In contrast, the sequential model recapitulates the bias in branch location from the older lattice near the minus-end and the resulting exclusion of the plus-ends (*Figure 2A–B*, green curves), as more nucleation sites deposit on the older lattice while the mother's plus-end continues to grow in the time it takes on average to nucleate a branch from the deposited site.

Is branched network architecture sensitive to the parameters in the sequential model? By changing the dimensionless ratio of the branching rate constant to the effective binding rate constant $\left( \frac{k_{branch}}{\sqrt{k_{bind}\, v_{pe}}} \right)$ derived in Appendix 1, we find that the bias is present when the branching rate is slower than the binding of nucleators (*Figure 2—figure supplement 2C*). Further decreasing the ratio of rates does not result in a more pronounced bias because the relative profile of deposited nucleation sites remains unchanged. However, when the second reaction step is fast and no longer rate-limiting (*Figure 2—figure supplement 2C*, $\left( \frac{k_{branch}}{\sqrt{k_{bind}\, v_{pe}}} \right) > 100$), the sequential model effectively approaches the single-step model. Thus, our model results in two regimes of network architectures and is not sensitive to chosen parameters.

Our simulations further show that the minus-end bias persists for every subsequent branching event in a dense network, similar to the first nucleation event, and broadly characterizes branched MT networks (compare *Figure 2—figure supplement 2D–E* with *Figure 2A–B*). In conclusion, the sequential reaction model recapitulates the spatial nucleation profile that we measured experimentally, whereas the single-step model does not.

## Architecture of branched networks at high microtubule density

To study how branched networks evolve over time, we next characterized the network architecture for large branched networks. By measuring the position of nucleation sites using our simulations, we observed that minus-ends are typically located closer to the origin of the network (defined as the minus-end of the first MT) in the sequential model than in the single-step model (*Figure 2C*). Measuring the position of plus-ends as well as the tubulin intensity shows that the total MT mass and plus-ends are also closer to the origin in the sequential model (*Figure 2D–E*).

Is either organization reflected in our experimental networks? The MT plus-ends and tubulin intensity profile in our experimental networks (*Figure 2—figure supplement 2*) are distributed closer to the origin (*Figure 2D–E*, blue), in striking agreement with the sequential model

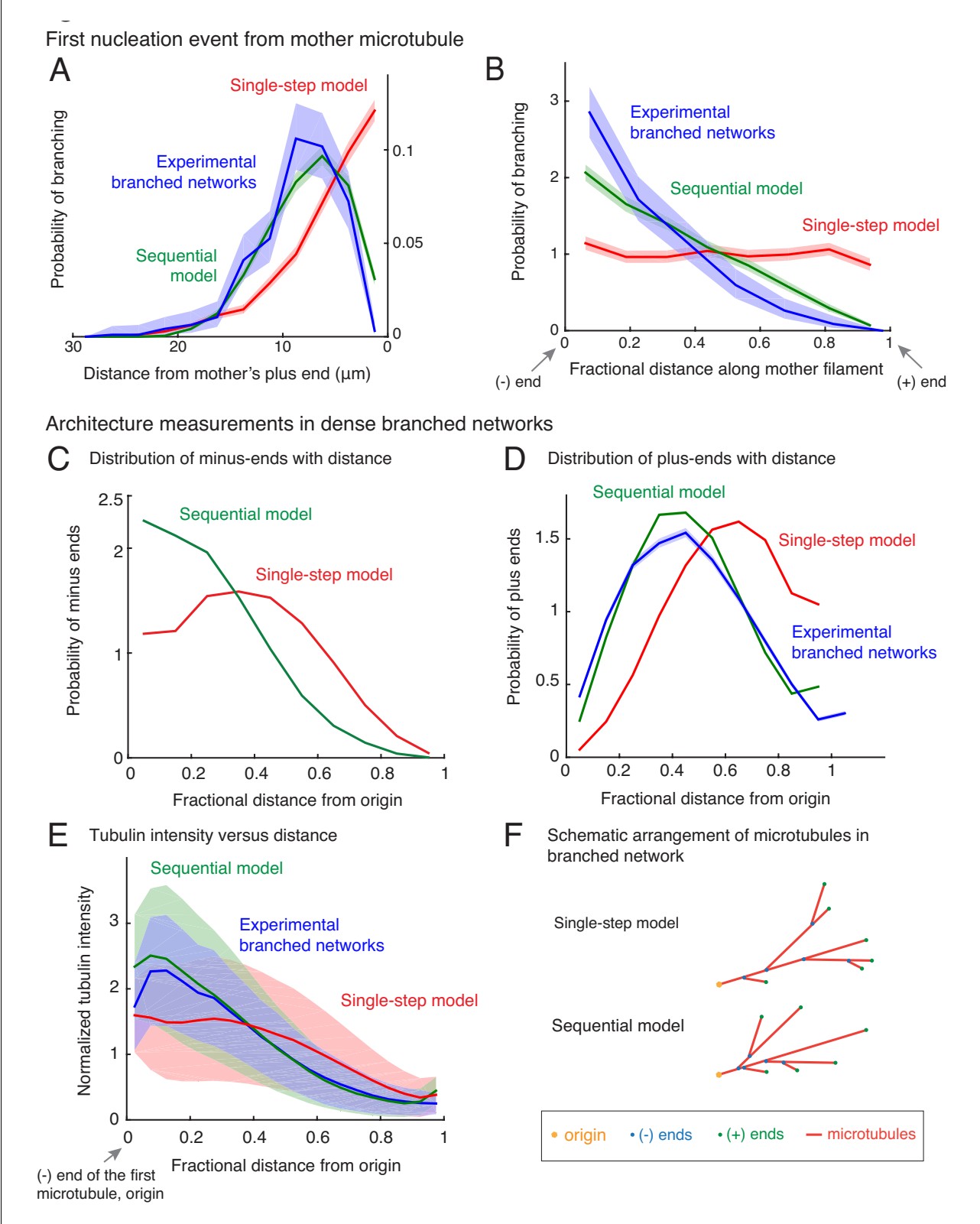

**Figure 2.** Nucleation profile of first branching event and spatial organization of microtubules in dense branched networks. Spatial location of first branching event on a naked mother MT was recorded during the formation of individual branched MT networks, and the following distributions were measured. See *Figure 2—figure supplement 1A* for representative examples. (**A**) Distance of the nucleation site from the mother's plus-end was measured at the time point when first branching event occurred. Rightmost (0 μm) point on the horizontal axis denotes nucleation near the mother's

*Figure 2 continued on next page*

*Figure 2 continued*

plus-end. Inverted x-axis is plotted for consistency with (B). (B) Fractional distance was obtained by dividing the branching nucleation site from the mother's minus-end by the total length of the mother MT when the first branching event occurred. Leftmost (0) point on x-axis denotes nucleation near the mother's minus-end, while rightmost (1) represents nucleation near the mother's plus-end. For panels (A-B), normalized probability of branching was plotted for n = 381 experimental measurements (blue), and n = 4000 each using the single-step model (red) or the sequential model (green). Shaded regions depict the 95% bootstrap confidence intervals. The experiments and analyses were performed with three independent extract preparations and all data was pooled. See also *Figure 2—figure supplement 1*. Spatial organization of MTs in dense branched networks. (C) Positions of minus-ends were obtained in simulated branched networks. The distances of all minus-ends were calculated from the origin (seed MT's minus-end, panel (F)), then normalized by seed MT's length (set to 1). The resulting probability distribution of was plotted. Data was pooled for 10–60 MTs in roughly 250 branched networks each simulated using the single-step model (red, n = 633517) and sequential model (green, n = 495794). Shaded regions depict the 95% bootstrap confidence intervals. (D) Positions of MT plus-ends were obtained in branched networks. Distances of all plus-ends were calculated from the origin (see panel (F)), normalized by the largest distance (set to 1), and the resulting probability distribution was plotted. For experimental measurements, data was pooled from 27 branched networks, each containing 10–55 EB1 comets, generated in two independent extract preparations (blue, n = 47362). For simulated networks, data was pooled from 10 to 60 MTs in 250 branched networks each simulated using the single-step model (red, n = 633517) and sequential model (green, n = 495794). Shaded regions depict the 95% bootstrap confidence intervals. See *Figure 2—figure supplement 2B* for the measurement procedure. (E) Tubulin intensity in branched networks was plotted against distance from the origin, normalized by the longest distance (set to 1). For experimental data, measurements were pooled for 33 branched networks (blue) observed in one *Xenopus* extract preparation, and analysis was repeated with three independent extract preparations. For simulations, data was pooled for 20–50 MTs in approximately 250 branched networks simulated using the single-step model (red) or the sequential model (green). Shaded regions depict the standard deviation in intensity measurements. See *Figure 2—figure supplement 2A* for the measurement procedure. (F) Schematic representation of MT organization in branched networks generated via the single-step model (top) or the sequential model (bottom). While the total number of MTs and their lengths is the same in the two representations, the overall distribution of plus-ends, minus-ends and MT mass is more concentrated near the origin (yellow) in the sequential model. See also *Figure 2—figure supplement 3*.

DOI: https://doi.org/10.7554/eLife.43890.010

The following figure supplements are available for figure 2:

**Figure supplement 1.** Measurement of nucleation profile and testing robustness of sequential model.
DOI: https://doi.org/10.7554/eLife.43890.011
**Figure supplement 2.** Spatial organization of microtubules in dense branched networks.
DOI: https://doi.org/10.7554/eLife.43890.012
**Figure supplement 3.** Self-similarity in the architecture of branched networks.
DOI: https://doi.org/10.7554/eLife.43890.013

(*Figure 2D–E*, green). We conclude that the sequential model recapitulates the spatial architecture of branched networks, and branching results in origin-biased MT organization due to higher age of those MT lattice regions (*Figure 2F*).

Interestingly, networks simulated by the two models do not change their overall architecture over time such that the normalized minus-end, plus-end or tubulin intensity distributions remain unchanged with time as the MT networks grow (*Figure 2—figure supplement 3A–C*). This suggests that branched networks are self-similar in time and maintain their architecture, which is a key characteristic of fractal structures (*Meakin, 1990*). Can this feature be observed in our experimental networks too? Measuring these distributions experimentally at early and late time-points also shows evidence of self-similarity (*Figure 2—figure supplement 3D–E*). Taken together, our data suggests that branched MT networks are self-similar in time, where branching events occur via a sequential, autocatalytic reaction scheme.

## Concentration of TPX2 and augmin determine the branching nucleation profile

We next asked if the known branching effectors, TPX2 and augmin, participate in the rate-limiting steps of our sequential model (*Figure 3*). We reasoned that if the unimolecular binding rate of augmin or TPX2 to branched networks underlie the two steps in our sequential model, the net branching nucleation rate can be altered by proportionately varying the bulk concentration of these two effectors in *Xenopus* egg extracts. We first modified the concentration of TPX2 by immunodepleting it to 20% of the endogenous concentration or adding recombinant TPX2 protein at 9.6-fold excess (*Figure 3A*, *Figure 3—figure supplement 1A–B* and *Video 5*). In parallel, we partially immunodepleted the augmin complex to 20% of the endogenous concentration in *Xenopus* egg extracts (*Figure 3A*, *Figure 3—figure supplement 1C* and *Video 5*). We observed that branching was

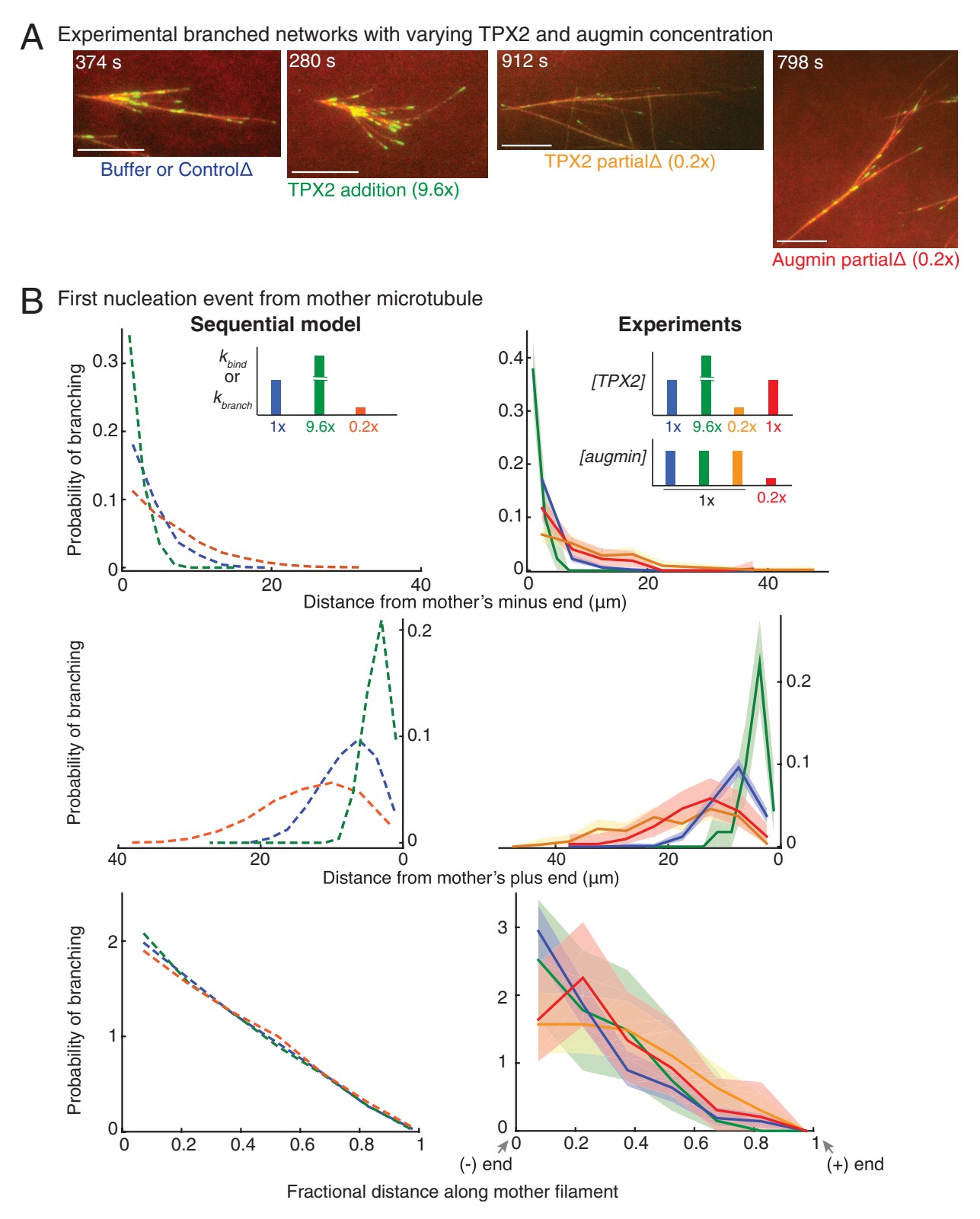

**Figure 3.** Network architecture changes with varying protein concentrations as predicted by the sequential model. (**A**) Branched MT networks were generated in *Xenopus* egg extracts with 10 µM RanQ69L for four conditions: buffer addition or control depletion, addition of 260 nM TPX2 (9.6x), partial depletion of TPX2 (20% of control), or partial depletion of augmin (20% of control). For each condition, at least 5–10 identical reactions were performed with each preparation of *Xenopus* extracts, and repeated with at least three independent extract preparations, except TPX2 addition was repeated with

*Figure 3 continued on next page*

*Figure 3 continued*

two independent extract preparations. Scale bars, 10 μm. See *Figure 3—figure supplement 1*. (B) Spatial location of first branching event on a naked mother MT was recorded for each condition in (A) and compared with prediction from sequential model. Distance of the branching nucleation site from the mother MT's minus-end (top), plus-end (middle) and fractional distance of nucleation from the minus-end (bottom) were measured, resulting probability distribution was plotted. Shaded area represents 95% confidence interval. Leftmost and rightmost points on x-axis denote nucleation site near the mother's minus-end or the plus-end respectively. The insets show the rate constants in sequential model corresponding to changes in protein concentrations: blue (1x rate constants, control reactions), green (9.6x $k_{bind}$ or $k_{branch}$, 9.6x TPX2 concentration), mustard and red (0.2x TPX2 and augmin concentration, respectively) compared to orange (0.2x $k_{bind}$ or $k_{branch}$). Number of experimental measurements: buffer or control depletion (blue, n = 283), TPX2 addition (green, n = 45), TPX2 partial depletion (yellow, n = 157), augmin partial depletion (red, n = 65). Number of simulations: n = 4000 for each condition. See *Figure 3—figure supplement 1* for example measurements and parameter fit for the model. See also *Figure 3—figure supplement 1*.

DOI: https://doi.org/10.7554/eLife.43890.014

The following figure supplement is available for figure 3:

**Figure supplement 1.** Controls for varying TPX2 and augmin concentration in *Xenopus* egg extracts.

DOI: https://doi.org/10.7554/eLife.43890.015

delayed with a decrease in concentration of augmin or TPX2 and accelerated with TPX2 addition (*Video 5*). Strikingly, TPX2 concentration determines the rate of nucleation in our networks (*Figure 3—figure supplement 1D*).

To quantitatively compare these experimental perturbations with the equivalent variation in rate constants, we measured the position of the first branching event on a mother MT (*Figure 3B*). The distance of the first branching site from the mother MT's minus- or plus-end increased with decreasing TPX2 or augmin to 20% endogenous concentration (red and yellow curves, *Figure 3B*, top and middle panels). This variation in the nucleation profile quantitatively agrees with the predictions from our model when the binding or branching rate constants, $k_{bind}$ and $k_{branch}$, were decreased to 20%, because the mother MTs grow longer before a branching event occurs (orange curve, *Figure 3B*, top and middle panels). Similarly, increasing the concentration of TPX2 to 9.6-fold decreases the distance between nucleation site and mother MT's minus- or plus-end, which quantitatively compares with equivalent increase in $k_{bind}$ or $k_{branch}$ (green curve, *Figure 3B*, top and middle panels). Finally, our model predicts that the non-dimensional spatial bias remains unaltered upon varying the rate constants by 5–10 fold, which was reproduced in experimental profiles upon equivalent change in concentration of TPX2 or augmin (*Figure 3B*, bottom panel).

Altogether, we show that altering the number of TPX2 or augmin molecules in bulk cytoplasm matches equivalent variations in the rate-limiting steps of our sequential model. Thus, the unimolecular kinetics of TPX2 and augmin result in either the deposition of nucleation sites or the subsequent emergence of a branch.

## Branching microtubule nucleation occurs via a sequential mechanism involving first TPX2 followed by augmin/γ-TuRC

We next aimed to identify which reaction steps are governed by TPX2 and augmin in our sequential model through a series of two-color

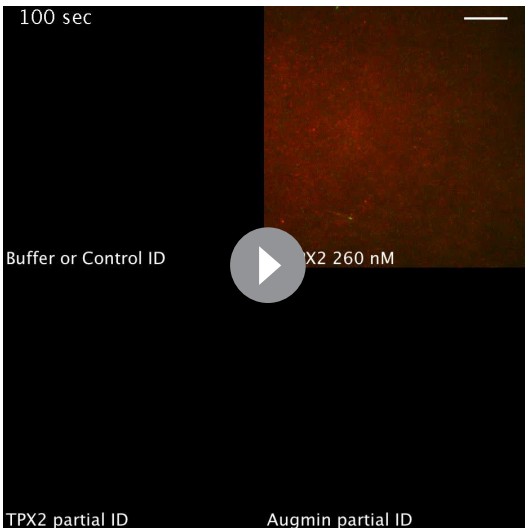

**Video 5.** Branched microtubule networks generated with varying augmin and TPX2 concentrations. Branched MT networks were generated in *Xenopus* egg extracts with 10 μM RanQ69L for four conditions: buffer addition or control depletion, addition of 260 nM TPX2, partial depletion of TPX2 (20% of control), or partial depletion of augmin (20% of control). MTs are labeled with Cy5-tubulin (red) and their plus-ends with EB1-mCherry (pseudo-colored as green). Scale bars, 10 μm. Representative movies are displayed, where timepoint 0 s represents the start of the reaction.

DOI: https://doi.org/10.7554/eLife.43890.016

solution exchange experiments. Based on existing research, we expected that augmin recruits γ-TuRC to pre-existing MTs before TPX2 stimulates branch generation (*Alfaro-Aco et al., 2017*; *Hsia et al., 2014*; *Song et al., 2018*; *Uehara et al., 2009*). We first generated MTs in *Xenopus* egg extracts lacking augmin but containing TPX2, which leads to the growth of de novo MTs, visualized with Alexa-488 tubulin, but no branches (*Figure 4—figure supplement 1A–B*), as described previously (*Petry et al., 2013*; *Thawani et al., 2018*). These de novo MTs were then subjected to egg extracts where augmin but no additional TPX2 is present, and new MTs were visualized with Cy5-labeled tubulin (*Figure 4A*). To our surprise, new branches (red MTs with EB1-labeled plus-ends in green) nucleated from the de novo MTs (blue) that were formed initially with TPX2 but without augmin (*Figure 4A* and *Video 6*). Interestingly, the first branches appeared as early as 10–30 s within the addition of augmin (*Figure 4—figure supplement 1B*). The reverse sequence, where de novo MTs were formed in the presence of augmin, but without TPX2, followed by a switch to extract with TPX2 but lacking augmin, did not result in branch formation (*Figure 4B* and *Video 6*). Only elongation of the plus-ends of de novo MTs (blue) or new de novo nucleations (red) were observed. These results demonstrate that branching MT nucleation is a sequential pathway where TPX2's deposition on the de novo MTs precedes the binding of augmin and subsequent branch formation, while additional soluble TPX2 is not required to generate branches.

Augmin and γ-TuRC have been shown to bind directly at low concentrations (*Song et al., 2018*). Based on this, we hypothesized that the MT nucleator, γ-TuRC, was involved later in this sequential pathway together with augmin. To test this, we performed experiments where the availability of γ-TuRC was regulated. We first generated MTs in the absence of augmin where TPX2 and γ-TuRC are available, and then exchanged the solution to extract where no γ-TuRC is present (*Figure 4C* and *Video 7*). Here, no branching occurred and MTs only elongated from their plus-ends at early time points. This demonstrates that γ-TuRC is essential together with augmin in the branching pathway after TPX2 binding. We note that rare branches were observed at later time points (*Figure 4C*, rightmost panel) due to residual γ-TuRC molecules after immunodepletion (*Figure 4—figure supplement 1A and B*, inset). In summary, our results establish that branching MT nucleation occurs via a sequence of events, namely first TPX2's binding to de novo MTs, followed by the binding of augmin/γ-TuRC prior to nucleation of branch MT from the mother MT.

## Measuring the TPX2 binding rate to microtubules in *Xenopus* egg extracts

To further assess TPX2's role in depositing the branch nucleation sites, we measured the binding rate of TPX2 to individual MTs in *Xenopus* egg extracts during formation of branched networks. We hypothesized that TPX2 binds to the MT lattice significantly below lattice saturation resulting in preference towards the older regions of the MT lattice near the minus-ends, analogous to the spatial profile of the first branching effector in our sequential model (*Figure 4—figure supplement 2A*). To directly measure the binding rate of TPX2, we adapted our assay to form branches on passivated coverslips to reduce fluorescent background, and replaced the endogenous TPX2 with an equivalent concentration of GFP-labelled TPX2 (*Figure 4D*, *Figure 3—figure supplement 1A* and *Video 8*). First, we observed that TPX2-GFP associated with individual MTs, including the first de novo MT prior to the formation of the first branched MT (*Figure 4D* and *Video 8*), supporting the molecular sequence during branch formation. The binding of TPX2 to MTs was slow, such that the plus-ends were mostly devoid of TPX2 signal, and a decreasing profile of TPX2 from minus- to plus-ends was observed (*Figure 4D*). To quantify the binding rate of TPX2, we measured the rate of increase in TPX2 intensity at individual pixels on the MT lattice (*Figure 4E*). The fluorescence intensity of single TPX2-GFP molecules was obtained from fluorescence photobleaching traces and used for normalization (*Figure 4—figure supplement 2B–C*). We calculated the binding rate of TPX2 to the MT lattice to be $0.4 \pm 0.2$ molecules $\mu m^{-1}$ $s^{-1}$ (n = 32 traces). This value lies within the same order of magnitude of our model parameter $k_{bind} = 0.1$ molecules $\mu m^{-1}$ $s^{-1}$ for deposition of nucleation sites. Notably, TPX2's measured binding rate is too slow to saturate the newly formed MT lattice, which presents approximately 250 inter-protofilament dimer sites per micron per second (*Zhang et al., 2017*). Hence, the slow binding kinetics of TPX2 results in its localization on the older lattice regions near the MT minus-ends, corresponding to its role in marking the nucleation sites in our sequential kinetic model.

Lastly, direct visualization of GFP-tagged recombinant augmin or fluorescent γ-TuRC antibody during branched network formation (*Figure 4—figure supplement 2D–E*) showed that augmin and

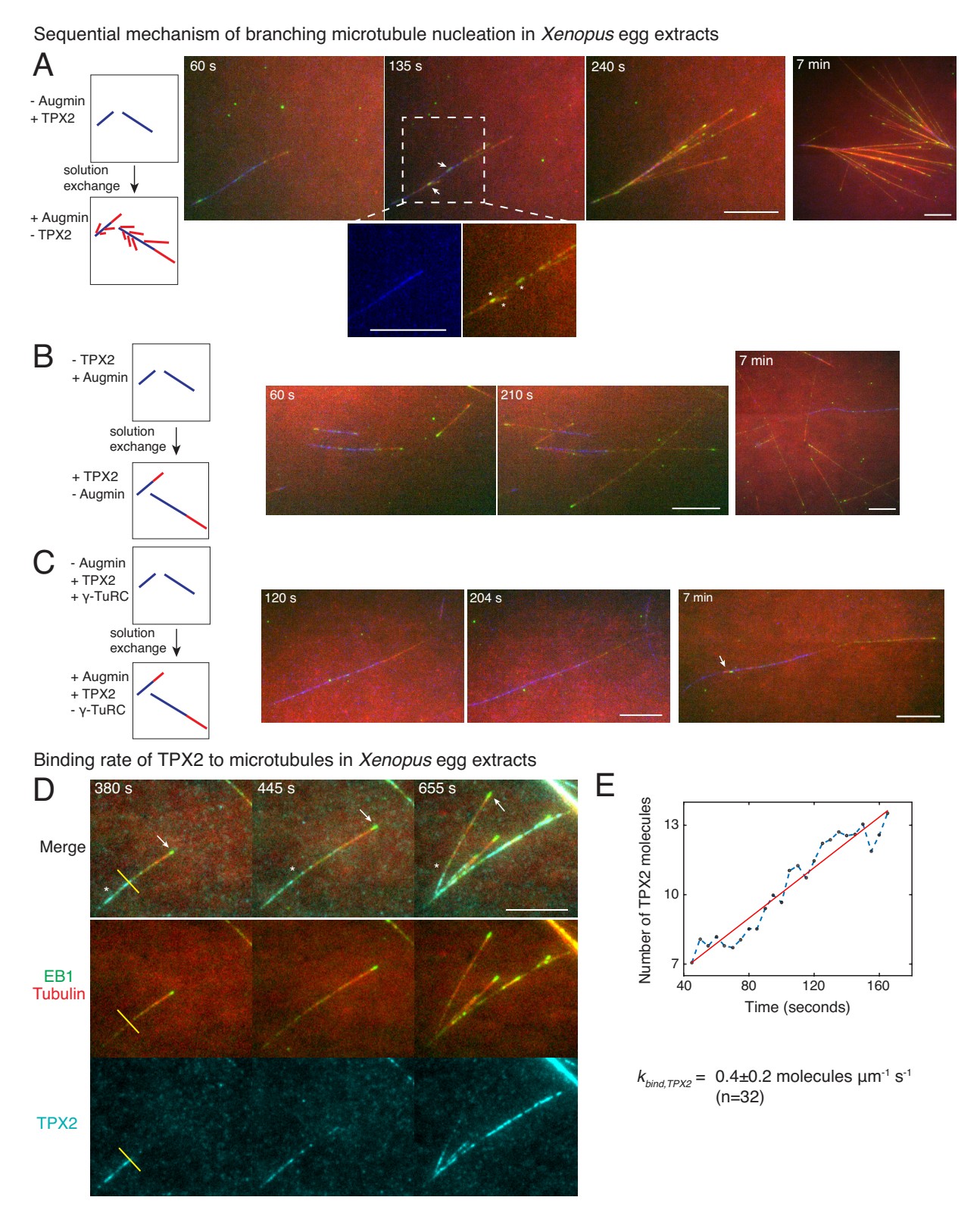

**Figure 4.** Sequential mechanism of branching microtubule nucleation and binding rate of TPX2 to microtubules. Sequential mechanism of branching MT nucleation (A-C). (A) De novo MTs (blue) were generated by performing branching reaction in augmin-depleted *Xenopus* egg extracts where TPX2 is present. Non MT-bound, soluble proteins were removed with buffer wash, and *Xenopus* egg extracts containing augmin but no TPX2 was introduced. Branched MTs (red), with their plus-ends labelled with EB1-mCherry (pseudo-colored as green), nucleated immediately from de novo MTs

*Figure 4 continued*

(blue), highlighted in the zoomed-in region. Late time point (7 min) shows formation of dense branched networks around the initial de novo MTs (blue). 0 s marks the time of extract exchange in the reaction chamber. Scale bar, 10 µm. The experiment was repeated six times with independent egg extract preparations. (B) De novo MTs (blue) were generated by performing branching reaction in *Xenopus* egg extracts containing augmin but no TPX2. Non MT-bound, soluble proteins were removed with buffer wash, and *Xenopus* egg extracts containing TPX2 but no augmin was introduced. No branching was seen, and only MT plus-ends elongated (Cy5-MTs in red and EB1-mCherry pseudo-colored as green) was observed. Late time point (7 min) depicted for comparison with (A). 0 s marks the time extract exchange in the reaction chamber. Scale bar, 10 µm. The experiment was repeated four times with independent egg extract preparations. (C) De novo MTs (blue) were generated by performing branching reaction in *Xenopus* egg extracts containing TPX2 and γ-TuRC but no augmin. Non MT-bound, soluble proteins were removed with buffer wash, and γ-TuRC-depleted *Xenopus* egg extracts containing TPX2 and augmin was introduced. At initial time points, only elongation of MT plus-ends (red with EB1-mCherry pseudo-colored as green) was observed. Rare branching events were seen at late time point (7 min), highlighted with a white arrow. 0 s marks the time extract exchange in the reaction chamber. Scale bar, 10 µm. The experiment was repeated thrice with independent extract preparations. See also *Figure 4—figure supplement 1*. (D-E) Endogenous TPX2 was replaced with 20–30 nM recombinant GFP-TPX2 in *Xenopus* egg extracts. Branched MT networks were generated with 10 µM RanQ69L, and time-lapse of TPX2 on the networks was recorded. MTs were labeled with Cy5-tubulin (red), their plus-ends with EB1 (green), and TPX2 is displayed in cyan. 0 s marks the start of the reaction. Scale bar, 10 µm. Arrows denote the plus-ends, while asterisks show TPX2's deposition on older lattice regions near the minus-ends and no binding to newly formed plus-ends. The experiment was repeated thrice with independent egg extract preparations. TPX2's intensity was measured over time on individual pixels (highlighted with a yellow line) corresponding to de novo MT before the first branching event occurred, normalized by single TPX2's fluorescence and converted into number TPX2 molecules. A representative trace is shown in (E), which was fit to a straight line (red). The slope was calculated to obtain the binding rate of TPX2 as $0.4 \pm 0.2$ (mean ± s.d.) molecules $\mu m^{-1}\ s^{-1}$ (n = 32 traces). A constant background noise level of 7 molecules was observed, which does not affect the calculated binding rate. The experiment was repeated thrice with independent extract preparations. See also *Figure 4—figure supplement 2*.
DOI: https://doi.org/10.7554/eLife.43890.017

The following figure supplements are available for figure 4:

**Figure supplement 1.** Control immunodepletion reactions and measurement of length of initial branched microtubules.
DOI: https://doi.org/10.7554/eLife.43890.018

**Figure supplement 2.** Controls for TPX2's binding rate measurement, and visualization of augmin/γ-TuRC on branched networks.
DOI: https://doi.org/10.7554/eLife.43890.019

γ-TuRC were not present initially on individual MTs but appear later as more MTs nucleate in the branched networks. This qualitatively supports the participation of augmin/γ-TuRC in the branch generation step. However, due to lower activity of recombinant augmin compared to the endogenous protein (*Song et al., 2018*), and indirect visualization of γ-TuRC, the quantitative measurement of this association rate may underestimate the kinetics of endogenous proteins. Thus, we cannot yet directly measure the association rate of endogenous augmin and γ-TuRC to branched networks. Altogether, our results demonstrate that branched MT networks are formed by a sequential reaction involving slow association of TPX2 to individual MTs, followed by binding of augmin/γ-TuRC molecules to generate a branched MT.

## Discussion

The pattern of MT nucleation that gives rise to the mitotic spindle in vivo remains to be determined because individual MTs cannot be resolved due to their high density in many regions of the cell (*Brugués et al., 2012*; *Burbank et al., 2006*; *Needleman et al., 2010*). Here, we study the formation of branched MT networks in purified cytoplasm as a model process that is essential for the assembly and maintenance of the spindle. Specifically, we investigated how key branching effectors build the MT nucleation profile and generate the branched network architecture. By combining precise experimental perturbations with image analysis tools, mathematical modeling, and detailed measurements, we show that nucleation in branched networks is spatially biased towards the older lattice regions near the MT minus-ends. We demonstrate that branching MT nucleation is a sequential process involving deposition of nucleation sites by TPX2 and delayed branching by the augmin/γ-TuRC complex. This sequential model comprehensively describes the branched network architecture. In sum, our study establishes how the dynamics of branching effectors generates the architecture of branched networks.

Our results provide new insights that rule out many alternative mechanisms of branching MT nucleation. In particular, mechanisms involving either only the direct localization of nucleators augmin/γ-TuRC to MTs, or the localization followed by the rate-limiting activation of augmin/γ-TuRC

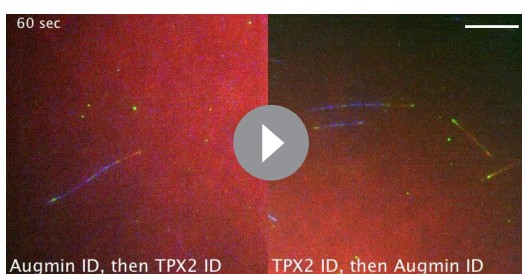

**Video 6.** Sequence of TPX2 and augmin in branching microtubule nucleation by two-color solution exchange experiments. Left movie: De novo MTs (blue) were generated in *Xenopus* egg extracts containing TPX2 but no augmin first, followed by exchange to *Xenopus* egg extracts containing augmin but no TPX2. Branched MTs labeled with Cy5-tubulin (red) and their plus-ends with EB1-mCherry (pseudo-colored as green) nucleate immediately from initial de novo MTs (blue) formed without augmin. Right movie: De novo MTs (blue) were generated in *Xenopus* egg extracts containing augmin but no TPX2 first, followed by exchange to *Xenopus* egg extracts containing TPX2 but no augmin. No branched MTs were observed, and only elongation of plus-ends labeled with Cy5-tubulin (red) and their plus-ends with EB1-mCherry (pseudo-colored as green) was seen. Reaction was performed 10 µM RanQ69L. Representative movies are displayed, where 0 s represents the time of solution exchange. Scale bars, 10 µm.

DOI: https://doi.org/10.7554/eLife.43890.020

(*Kollman et al., 2011*) do not result in branch formation (*Figure 4A–B*). Instead, TPX2's localization on individual MTs is necessary before augmin/γ-TuRC to nucleate a branch. Second, we show that branching MT nucleation is not a result of independent binding of TPX2 and augmin to the MT lattice, an activity that has been shown for both proteins in vitro (*Brunet et al., 2004*; *Hsia et al., 2014*; *Roostalu et al., 2015*; *Song et al., 2018*). Based on our results (*Figure 4A–B*), we speculate that localization of augmin to MTs in the cytoplasm is mediated by other proteins, as suggested recently for human augmin (*Luo et al., 2019*), and potentially by TPX2. Third, we find that the nucleator γ-TuRC cannot be recruited by TPX2 independently, but requires augmin (*Figure 4C*), which could not be ruled out based on previous work. Finally, we note that our work does not rule out the participation of other, yet unknown effectors in branching MT nucleation. The minor differences between our experimental nucleation profiles and the sequential model (*Figure 2*) may indicate the involvement of undiscovered effectors, or time-dependent regulation of known effectors. In the future, it will be important to identify new branching effectors and elucidate the detailed mechanism of branching with bottom-up in vitro reconstitution.

The nucleation and dynamics of branched MT networks recapitulate the key properties of the mitotic spindle measured previously. The length distribution of MTs in branched networks displays a bounded growth regime, as reported for centrosomes in mitosis (*Verde et al., 1992*).

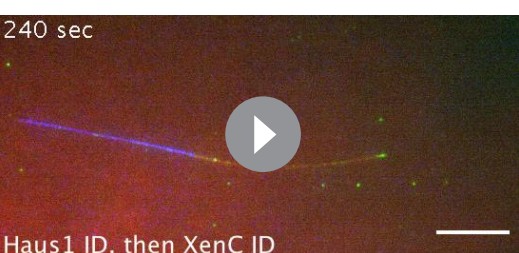

**Video 7.** Sequence of γ-TuRC, TPX2 and augmin in branching microtubule nucleation by two-color solution exchange experiments. De novo MTs (blue) were generated in *Xenopus* egg extracts containing TPX2 and γ-TuRC but no augmin first, followed by exchange to *Xenopus* egg extracts containing TPX2 and augmin but no γ-TuRC. No branched MTs were observed, and only elongation of plus-ends labeled with Cy5-tubulin (red) and their plus-ends with EB1-mCherry (pseudo-colored as green) was seen. Reaction was performed 10 µM RanQ69L. Representative movies are displayed, where 0 s represents the time of solution exchange. Scale bars, 10 µm.

DOI: https://doi.org/10.7554/eLife.43890.021

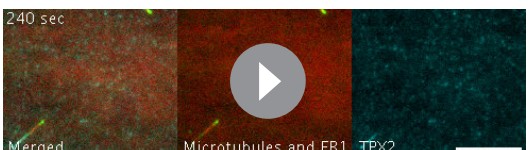

**Video 8.** Binding of TPX2 to branched microtubule networks in *Xenopus* egg extracts. Endogenous TPX2 was immunodepleted from *Xenopus* egg extracts and replaced with 20–30 nM recombinant GFP-TPX2. Branched MT networks were generated 10 µM RanQ69L, and TPX2's association with MTs was observed over time (cyan). MTs are labeled with Cy5-tubulin (red) and their plus-ends with EB1-mCherry (pseudo-colored as green). TPX2 signal was observed on older MT lattice near the minus-ends, and not on the newly formed lattice near the plus-ends. Representative movie is displayed, where time 0 s represents the start of the reaction. Scale bar, 10 µm.

DOI: https://doi.org/10.7554/eLife.43890.022

Thus, branched networks are sustained through new MT nucleation events. This is similar to recent models for spindle assembly and reflects the measured length distributions of spindle MTs (*Brugués et al., 2012*; *Needleman et al., 2010*). The net branching nucleation rate we directly measured (0.5 min$^{-1}$) agrees with previous estimates from monopolar spindles (3 min$^{-1}$; *Decker et al., 2018*) and RanGTP asters (1.2 min$^{-1}$; *Clausen and Ribbeck, 2007*).

Building on previous work, our study advances the current understanding of how MT nucleation builds the mitotic spindle. First, because of challenges associated with studying MT nucleation in the spindle due to its high MT density, many assumptions regarding autocatalytic nucleation needed to be made in previous works, such as autocatalytic nucleation occurs via a single MT binding event (*Clausen and Ribbeck, 2007*; *Loughlin et al., 2010*), which is encompassed by our single-step model and does not explain our measurements that show preferential nucleation from older MT regions. Second, because most measurements in the spindle are performed at metaphase after the spindle has assembled (*Brugués et al., 2012*; *Burbank et al., 2006*; *Needleman et al., 2010*; *Yang et al., 2007*), they lack the dynamic information needed to capture the multiple rate-limiting steps for branching MT nucleation (*Decker et al., 2018*; *Kaye et al., 2018*; *Oh et al., 2016*; *Oriola et al., 2018*), and thus distinguish between the single-step model (where branching rate is not limiting) and the sequential model (where branching rate limits the net reaction). Here, by directly probing the temporal dynamics of network assembly and time-resolved experiments, we show that the MT networks are organized by rate-limited deposition of nucleation sites and followed by branch emergence through binding of MT nucleators. We demonstrate that the localization of the nucleator γ-TuRC to spindle MTs regulates nucleation in the mitotic spindle. Third, and most importantly, the preferential nucleation from older MT lattice regions could allow for precise control over where branching occurs in the spindle. We propose that TPX2, which is released in the vicinity of chromosomes (*Kalab et al., 2002*; *Oh et al., 2016*), accumulates on the older MT lattice regions near chromosomes such as on MTs captured and stabilized by the kinetochores (*Cimini et al., 2003*; *DeLuca et al., 2011*; *Lampson et al., 2004*; *Umbreit et al., 2012*) and older lattice regions such as minus-ends near the spindle center. Branching MT nucleation by augmin/γ-TuRC from deposited TPX2 sites will selectively amplify older MTs generating dense kinetochore fibers near chromosomes.

The mitotic spindle is organized by regulating nucleation of MTs and their further sorting by molecular motors (*Brugués et al., 2012*; *Oriola et al., 2018*). With the new understanding of how the dynamics of molecular effectors organize autocatalytic MT nucleation in space and time, we can now begin to understand how the spindle architecture is constructed by spatially-regulated MT generation and motor-based active transport.

## Materials and methods

### Cloning, expression and purification of proteins

His-RanQ69L plasmid was a gift from J. Wilbur and R. Heald (*Weis et al., 1996*). N-terminal StrepII tag-6xHis-TEV mTagBFP2 RanQ69L was cloned into pST50Tr-STRHISNDHFR vector (*Tan et al., 2005*) using Gibson Assembly (New England Biolabs). All constructs used in this study were expressed in *E. coli* Rosetta2 cells (EMD Millipore) for 12–18 hr at 16°C or 7 hr at 25°C. For purification of His-RanQ69L, cells were lysed using an Emulsiflex (Avestin) in the binding buffer (50 mM NaPO$_4$, 600 mM NaCl, 20 mM Imidazole, 2.5 mM PMSF, 6 mM BME, pH 8.75) with additional protease inhibitor cocktail (Sigma) and DNAseI (Roche). Lysate was loaded on to HisTrap HP (GE Healthcare), His-RanQ69L was eluted in binding buffer with 500 mM Imidazole, and dialyzed into CSF-XB buffer (100 mM KCl, 10 mM K-HEPES, 5 mM K-EGTA, 1 mM MgCl$_2$, 0.1 mM CaCl$_2$, 10% w/v sucrose, pH 7.7). For purification of BFP-RanQ69L, cells were lysed in binding buffer (100 mM Tris-HCl, 450 mM NaCl, 1 mM MgCl$_2$, 1 mM EDTA, 2.5 mM PMSF, 6 mM BME, pH 8.75), loaded onto StrepTrap HP column (GE Healthcare), protein eluted in binding buffer with 2.5 mM D-desthiobiotin and dialyzed into CSF-XB buffer. 200 μM GTP was included in the lysis, elution and dialysis buffers for all RanQ69L purifications.

C-terminal GFP was replaced with mCherry tag in the pET21a vector carrying EB1 (*Petry et al., 2011*). EB1-mCherry was purified using His-affinity with binding buffer (50 mM NaPO$_4$, 500 mM NaCl, 20 mM Imidazole, 2.5 mM PMSF, 6 mM BME, pH 7.4) and eluted with 500 mM Imidazole from

HisTrap HP column. Peak fractions were pooled and loaded onto Superdex 200 pg 16/600 (GE Healthcare) and gel filtration was performed in CSF-XB buffer.

Full-length TPX2 with N-terminal Strep II-6xHis-GFP-TEV site tags was cloned into pST50 vector and expressed in bacteria for 7 hr at 25°C. TPX2 was first affinity purified using Ni-NTA beads in binding buffer (50 mM Tris-HCl pH 8.0, 750 mM NaCl, 15 mM Imidazole, 2.5 mM PMSF, 6 mM BME) and eluted with 200 mM Imidazole. All protein was pooled and diluted 4-fold to 200 mM final NaCl. Nucleotides were removed with a Heparin column (HiTrap Heparin HP, GE Healthcare) by binding protein in 250 mM NaCl and isocratic elution in 750 mM NaCl, all solutions prepared in Heparin buffer (50 mM Tris-HCl, pH 8.0, 2.5 mM PMSF, 6 mM BME). Peak fractions were pooled and loaded on to Superdex 200 pg 16/600, and gel filtration was performed in CSF-XB buffer.

TPX2 fragment containing amino-acids 320–631 was cloned in pGEX4T1 expression vector with an N-terminal GST tag, as described previously (*Alfaro-Aco et al., 2017*) and expressed in bacteria for 16 hr at 16°C. Cell pellet was lysed in binding buffer (50 mM Tris-HCl pH 8.0, 150 mM NaCl, 2.5 mM PMSF, 6 mM BME), bound to Glutathione agarose resin (ThermoScientific, catalog 25237), and eluted with 20 mM reduced L-Glutathione (Sigma, G4251). Peak fractions were pooled, loaded on to Superdex 200 10/300 and gel filtration was performed in CSF-XB buffer.

Porcine and bovine brain tubulin was labelled with Cy5- or Alexa-488 NHS ester (GE Healthcare) to 54% labelling using previously described methods (*Gell et al., 2010*).

All proteins were dialyzed or gel filtered into CSF-XB buffer (100 mM KCl, 10 mM K-HEPES, 5 mM K-EGTA, 1 mM MgCl$_2$, 0.1 mM CaCl$_2$, pH 7.7, 10% w/v sucrose), flash-frozen and stored at $-80°$C. Protein concentrations were determined with Bradford dye (Bio-Rad) and a Coomassie-stained SDS-PAGE against known concentrations of BSA (A7906, Sigma).

## Generation of antibodies and western blot analysis

Polyclonal XenC antibody was a gift from C. Wiese (*Wiese and Zheng, 2000*) and was used for γ-TuRC immunodepletion. For immunodepletion and western blot analysis of TPX2, rabbit polyclonal antibodies were generated (Yenzyme) against GST-tagged *Xenopus laevis* TPX2 α3–7 protein described previously (*Thawani et al., 2018*). For immunodepletion of augmin, His-tagged *Xenopus laevis* HAUS8 was used to produce rabbit polyclonal anti-HAUS8 anti-serum (Genscript). Alexa-647 labelled XenC antibody was generated by first dialyzing antibodies in PBS buffer (50 mM NaPO$_4$, 150 mM NaCl, pH 7.4), reaction with Alexa fluor 647 NHS ester (ThermoScientific, catalog A37573), and removal of unreacted dye by gel filtration in Bio-Gel P-30 Gel (Bio-Rad, catalog 1504150). Each molecule of XenC antibody was labelled with 2.5 Alexa-647 dye on average.

For western blot analysis of augmin subunits, anti-sera were generated against His-tagged HAUS1 and C-terminal fragment HAUS6 as well (Genscript). All custom-made antibodies were purified from serum with an antigen-coupled matrix (Affi-Gel 10 or 15, Biorad). For western blot analysis of other proteins, mouse anti-γ-tubulin (GTU88, Sigma, 1:3000 dilution), mouse anti-GCP4 (D-5, Santa Cruz Biotechnology, 1:100 dilution), mouse anti-α-tubulin (T5168, Sigma, 1:5000 dilution) antibodies were used. For all western blots in *Figure 3* and related supplements, quantitative, fluorescent western blotting was performed with anti-rabbit or mouse secondary antibodies (IRdye 800 nm, LI-COR, 1:10000 dilution) and imaged with Odyssey CLx imaging station (LI-COR). Western blot analysis of *Figure 4—figure supplement 1A* was performed with chemiluminescence.

## Preparation of *Xenopus* egg extracts and immunodepletion experiments

CSF extracts were prepared from *Xenopus laevis* oocytes as described previously and either used immediately or immunodepleted (*Petry et al., 2011*; *Thawani et al., 2018*). For partial immunodepletions, TPX2-, HAUS8 (MT binding subunit of augmin complex)- and rabbit IgG(control)-antibody were conjugated to Protein A Dynabeads (Life Technologies #10002D) overnight. For complete immunodepletion of augmin or γ-TuRC, HAUS1 and XenC antibodies were used, respectively. The amount of antibody coupled and incubation time for partial immunodepletion of proteins were determined empirically. For 80% depletion of endogenous TPX2, 14 μg of purified anti-TPX2 antibody (0.14 mg/ml purified stock) was coupled with 100 μl Dynabeads overnight and was incubated with 100 μl egg extract for 20 min. For complete TPX2, augmin or γ-TuRC immunodepletion, 28 μg of purified anti-TPX2, HAUS1- or XenC- antibody was coupled to 200 μl Dynabeads overnight and

100 µl egg extract was depleted in two rounds by incubating with 1:1 bead volume for 40–50 min in each round. For 80% depletion of endogenous augmin complex, 4 µg of purified anti-HAUS8 antibody (1.44 mg/ml stock) was coupled with 100 µl Dynabeads overnight and was incubated with 100 µl egg extract for 20 min. Control immunodepletion was always performed with rabbit IgG antibody and depletion efficiency was assessed using western blots and functional assays.

## Branching microtubule nucleation and data acquisition for architecture measurements

Microtubule (MT) nucleation was assayed in *Xenopus* egg extracts as described recently (*Thawani et al., 2018*). Briefly, fresh CSF extracts were prepared and subjected to immunodepletion. Channels were prepared between glass slides (Fisher Scientific) and 22 × 22 mm, no 1.5 coverslips (12-541B, Fisherbrand) using double-sided sticky tape. All reactions were performed with 0.5 mM vanadate (sodium orthovanadate, NEB) to avoid sliding of MTs on the coverslips. The following reaction was prepared: 10 µl reaction containing 7.5 µl *Xenopus* egg extracts, 10 µM BFP- or his-tagged RanQ69L, 0.89 µM Cy5-labeled tubulin, 200 nM EB1-mCherry and 0.5 mM vanadate. For excess recombinant TPX2 addition, the specified concentration of GFP-TPX2 was also added to the reaction. All proteins and chemicals added to egg extracts were stored or diluted into CSF-XB buffer and no more than 25% dilution of extract was performed for any reaction. The reaction was mixed and incubated on ice for 2–3 min to reduce the effect of handing, and 6 µl mixture was pipetted into a flow cell to start the reaction.

The temperature switch from ice to ambient condition allows for MT nucleation to occur and reaction was imaged immediately to capture the initial phase of assembly of branched networks. Each nucleation reaction was started by incubating egg extract at 18°C in a temperature-controlled room. This also marked the start of the reaction, 0 s, unless specified otherwise. Total internal reflection fluorescence (TIRF) microscopy was performed with Nikon TiE microscope using 100 × 1.49 NA objective. Andor Zyla sCMOS camera was used for acquisition, with a large field of view of 165.1 × 139.3 µm. 2 × 2 binned, dual color images were acquired using NIS-Elements software (Nikon). Brightness and contrast were optimized individually for display.

For experimental measurements of wild-type branched MT networks reported in *Figures 1* and *2A–B* and related supplementary panels, EB1 and tubulin channel were imaged as branched networks in each reaction at 0.5 frames per second for 5–15 min. For plus-end density measurements in *Figure 2C* and *Figure 2—figure supplements 2–3*, only EB1 channel was imaged at 0.5 frames per second for 5–15 min. For tubulin intensity measurements in *Figure 2D* and *Figure 2—figure supplements 2–3*, only single snapshots of tubulin and EB1 channels were recorded to avoid fluorescence photobleaching, which would alter the fluorescence measurements. In this case, for each prepared reaction, multiple fields of view containing multiple branched networks were imaged at a single time point using multi-XY acquisition in NIS-Elements, while two time points (early and late) were imaged for experiment shown in *Figure 2—figure supplement 3E*. For experiments shown in *Figure 3* and related supplements, EB1 and tubulin channels were imaged in immunodepleted and control extracts at 0.5 frames per second. For all measurements performed in this study, the same reaction had to be repeated between 8–10 times with each *Xenopus* egg extract to obtain large number of measurements and build probability distributions. Particularly, to compare the architecture between immunodepletions or protein additions with control (*Figure 3*), the control and experimental reactions were each performed 8–10 times alternately with every *Xenopus* egg extract preparation.

## Two-color solution exchange experiments in *Xenopus* egg extracts

18 × 18 mm glass coverslips were coated with dichlorodimethylsilane (*Gell et al., 2010*) and used within a week of preparation. First, a fragment of TPX2 with amino-acids 320–631, which neither induces branching MT nucleation nor functions as dominant negative (*Alfaro-Aco et al., 2017*), was attached to silane-coverslips at 4 µM concentration to bring down MTs generated *Xenopus* egg extracts onto the imaging field. Unattached protein was rinsed with BRB80 buffer (80 mM K-PIPES, 1 mM MgCl$_2$, 1 mM K-EGTA, pH 6.8), coverslip was blocked with 2.5 mg/ml κ-casein (Sigma, catalog C0406), rinsed with BRB80 buffer then further rinsed with CSF-XB buffer. This prepared reaction chamber was used for the two-color solution exchange experiment with *Xenopus* egg extracts.

Augmin, TPX2 and γ-TuRC were first individually immunodepleted from *Xenopus* egg extracts. For experiments shown in *Figure 4A* and *Figure 4—figure supplements 1B*, 10 µl reaction containing 7.5 µl augmin-depleted extract, 10 µM RanQ69L, 0.89 µM Alexa-488-labeled tubulin, 200 nM EB1-mCherry and 0.5 mM vanadate was first added to the reaction chamber. Individual, de novo MTs were allowed to form for 10 min, and soluble, non-MT bound proteins were removed by washing with 20 µl CSF-XB. 10 µl reaction containing 7.5 µl TPX2-depleted extract, 10 µM RanQ69L, 0.89 µM Cy5-labeled tubulin, 200 nM EB1-mCherry and 0.5 mM vanadate was then introduced and imaged immediately. De novo MTs in the center of the channel were imaged where maximum solution exchange occurs, and not near the edges of the channel to avoid boundary effects. For *Figure 4—figure supplement 1B*, TPX2-depleted extract was introduced in the chamber while imaging the de novo MTs to measure the time difference between solution exchange and generation of branches. For experiment shown in *Figure 4B*, 10 µl reaction containing 7.5 µl TPX2-depleted extract, 10 µM RanQ69L, 0.89 µM Alexa-488-labeled tubulin, 200 nM EB1-mCherry and 0.5 mM vanadate was first added to the reaction chamber. Individual, de novo MTs were allowed to form for 10 min, and soluble, non-MT bound proteins were removed by washing with 20 µl CSF-XB. 10 µl reaction containing 7.5 µl augmin-depleted extract, 10 µM RanQ69L, 0.89 µM Cy5-labeled tubulin, 200 nM EB1-mCherry and 0.5 mM vanadate was then introduced and imaged immediately. For experiment shown in *Figure 4C*, 10 µl reaction containing 7.5 µl augmin-depleted extract, 10 µM RanQ69L, 0.89 µM Alexa-488-labeled tubulin, 200 nM EB1-mCherry and 0.5 mM vanadate was first added to the reaction chamber. Individual, de novo MTs were allowed to form for 10 min, and soluble, non-MT bound proteins were removed by washing with 20 µl CSF-XB. 10 µl reaction containing 7.5 µl γ-TuRC-depleted extract, 10 µM RanQ69L, 0.89 µM Cy5-labeled tubulin, 200 nM EB1-mCherry and 0.5 mM vanadate was then introduced and imaged immediately. Time 0 s denotes the exchange of final solution in these experiments.

## Visualization of TPX2, augmin and γ-TuRC on branched microtubule networks and measurement of binding rate of TPX2

The reaction chamber was prepared with silane-passivated coverslips, as described above, and used for visualization of fluorescent proteins during the formation of branched MT networks.

For observing TPX2, endogenous TPX2 was immunodepleted, 25 nM recombinant GFP-TPX2 was added, and incubated in *Xenopus* egg extracts for 20 min. Following this, 10 µM RanQ69L, 0.89 µM Alexa-488-labeled tubulin, 200 nM EB1-mCherry and 0.5 mM vanadate were added and branched MT networks were imaged in the reaction chamber. EB1, tubulin and TPX2 channels were imaged at 0.2 frames per second. For observing augmin or γ-TuRC, 30 nM of GFP-tagged augmin holocomplex (*Song et al., 2018*) or 2 µg/ml of Alexa-647 labeled XenC antibody was added to mock-depleted *Xenopus* egg extracts in the presence of 10 µM RanQ69L, 0.89 µM Cy5- or Alexa-488-labeled tubulin, 200 nM EB1-mCherry and 0.5 mM vanadate. Branched MT networks were formed in the reaction chamber. EB1, tubulin and augmin or γ-TuRC channels were imaged at 0.28 frames per second. To observe fluorescent molecules with high signal to noise, Andor iXon 897 EMCCD camera was used for acquisition, with an EM-gain of 300 and field of view of 81.3 × 81.3 µm.

Binding rate of TPX2 to individual MTs during branch network formation was measured on individual, de novo MT before the nucleation of first branched MT as follows. TPX2's image sequence was isolated and plotted across time using the 'reslice' function in ImageJ. TPX2 intensity versus time traces for individual pixels corresponding to the MT lattice were obtained, and their linear regions were fit to first-order polynomial using MATLAB curve fit. Slope of the fit gives TPX2's binding rate in intensity pixel$^{-1}$ second$^{-1}$, which was then converted into molecules µm$^{-1}$ s$^{-1}$ using the known pixel size of 0.16 µm and measured intensity of per TPX2 molecule, as described below.

To obtain fluorescence intensity of single TPX2 molecules, photobleaching of GFP-TPX2 was performed after non-specifically adhering TPX2 to glass coverslips. The timescale of photobleaching was obtained from the trace of mean intensity over time (*Figure 4—figure supplement 2B*), which was much longer than the number of frames used for measuring binding rate. Bleaching events of individual TPX2 molecules were found and the peak intensity versus time trace was fit to Heaviside function in MATLAB (*Figure 4—figure supplement 2B*). Fluorescence intensity of single molecules was measured from the fit for n = 10 traces as 2700 ± 300 au. Photobleaching and binding measurements of TPX2 were performed in the same experimental set with identical imaging and camera settings.

## Tracking microtubules in branched networks

Individual branched MT networks were cropped in the image sequence and all MTs were tracked using the following steps.

### Detection and tracking of EB1 comets

EB1 comets on the MT plus-ends were detected and tracked as follows. First, EB1 channel was pre-processed using a moving median filter of 15 frames, using the source MATLAB file *movingmedian.m* in *Supplementary file 1*, where for each pixel in every time frame of the image sequence, the median intensity of 7 frames before and after the specific time frame is subtracted from the intensity in the specific frame (step one in *Figure 1—figure supplement 1*). The moving median filter selects for moving particles in the field of view, which are the EB1 comets for our datasets. This pre-filtered EB1 channel is input to the comet detection and plus-end tracking modules in the open source, *uTrack* v2.1.3 software (step two in *Figure 1—figure supplement 1*; *Applegate et al., 2011*; *Jaqaman et al., 2008*). These modules were accessed via the MATLAB scripts *script_preprocess_runuTrack.m* and *runUTrack.m* in *Supplementary file 1*. The detailed description on these modules is provided in the *uTrack* documentation. For this work, the pixel size (130 nm), frame interval (2 s), numerical aperture (1.49 NA), fluorophore excitation and emission wavelength (561 nm and 620 nm, respectively), exposure time (between 100–200 ms) were specified. Due to the size of EB1 comets (length ≈ 10 pixels, width ≈ 4 pixels), the band-pass filter with the high-pass Gaussian standard deviation (s.d.) of 14 pixels and the low-pass Gaussian s.d. of 1 pixel were applied. For watershed segmentation, the parameters used were varied so that most of the EB1 comets were detected, while the background noise and false-positive detections were minimized. A threshold of 12–18 s.d. in intensity and a step-size of 1–6 s.d. was chosen. The parameters were held constant, for analyzing an entire dataset of branched networks imaged with one *Xenopus* egg extract preparation as much as possible, and identical image acquisition settings for used for the entire dataset. The parameters were optimized further if errors were detected. For tracking EB1 comets, the MT plus-end tracking module of *uTrack* was used, which performs linking of comets in consecutive frames (frame-to-frame linking), and short gaps of up to 5 frames between track segments were closed. The following parameters were specified in *uTrack* and detailed description of the parameters is provided documentation supporting *uTrack* v2.1.3.

| Parameters | Values |
| --- | --- |
| Time window for gap closing | 5 frames |
| Minimum track segment length | 3 frames |
| Minimum search radius | 0 pixels |
| Maximum search radius | 9 pixels |
| Brownian search radius (multiplication factor) | 2 |
| Maximum forward angle for gap closing | 20 degrees |
| Fluctuation radius | 4 pixels |
| Kalman function search radius for the first iteration | 7 pixels |

The expansion of search radius was not chosen during tracking, non-linear tracks were accepted and not broken, and catastrophes were not detected by setting backward angle permitted as well as shrinking velocity to zero, because a low frequency of catastrophe events were in our assay setup. Finally, we obtain EB1 comet tracks from this procedure.

### Merging EB1 tracks to obtain complete microtubule trajectory

The code used here is provided in *Supplementary file 1* as *mergeEBTracks.m* with accompanying functions *calculateijDirection.m*, *findTrackInfoStartsEnds.m*, *lineIntegral.m* and *calculatePerpDistance.m*.

## Overview

To reconstruct the entire MT trajectories that form the branched networks, EB1 tracks were merged using a custom temporally greedy optimization procedure. The greedy optimization described below was obtained empirically based on the nature of EB1 track breaks observed in our experimental data. We note that the global optimization for merging as implemented in *uTrack* (*Applegate et al., 2011*) was attempted and found to be unfit for our dataset. This is because branching yields many EB1 tracks that are roughly in the same direction and location as a result of shallow branch angle and positional bias respectively, and global optimization produced many false merges. Therefore, a temporally greedy optimization with custom criteria was found to be a more successful approach. For this study, the below described procedure successfully reconstructs entire trajectories of up to 40 MTs over the duration of tens of minutes with few errors.

## Microtubule intensity calculation

MT intensity in the gaps was also used to ensure the correct gap merging. For this, the line integral of tubulin intensity $I_{MT}(i,j)$ was calculated for every candidate the gap between EB1 tracks $i$ and $j$ as follows. The tubulin channel imaged was first processed in ImageJ to correct for background illumination and bleaching. Background illumination was corrected by filtering out large variations using Bandpass filter under Process, FFT. Large structures of 60 pixels were filtered out. Bleaching was corrected by first selecting a background region where MTs do not emerge. Z-axis profile of this ROI was plotted and these values were fit to an exponential curve with offset using curve-fitting. The intercept value 'c' was recorded. Simple ratio bleach correction was selected and 'c' value was used as background. Finally, the first frame of the output, where no MTs have appeared yet, was subtracted from the rest of stack. This normalizes the tubulin intensity for varying background level between imaging conditions and makes the following code robust.

Tubulin line integral $I_{MT}(i,j)$ was calculated between end position of track $i$ ($p_{end,i}$) and start position of track $j$ ($p_{start,j}$). The cartesian distance between $p_{end,i}$ and $p_{start,j}$ was divided into points spaced by individual pixel length. Image intensity was interpolated at each of these points and summed to obtain the line integral. To compare this value for all candidates, the line integral was divided by the cartesian distance ($|p_{end,i} - p_{start,j}|$) and assigned to $I_{MT}(i,j)$.

The cutoff tubulin intensity $I_{cutoff}$ was determined for each stack as follows. For one of our experimental datasets, we made the merge assignments manually and found the median gap length to be 8.8 pixels. $I_{cutoff}$ was set to 20 x mean intensity from the processed tubulin channel/median gap length.

## Specific conditions

First, short, spurious EB1 tracks detected were filtered out based on their length and net displacement. Tracks smaller than 2–4 frames with net displacement smaller than two pixels or net speed lower than one pixel per frame were discarded. The rest of the EB1 tracks were considered for the following merge procedure.

A maximum time frame interval ($\Delta t_{max}$ = 30–40 frames) was defined for any possible merge between the end of track $i$ and start of candidate track $j$. For each track pair to be merged, we record five quantities in a matrix form: number of gap frames between the two tracks, $t_{gap}(i,j)$; the perpendicular distance between the two tracks, $D_{perp}(i,j)$; the dot product between the two track orientations, $N_{dot}(i,j)$; the minimum distance recorded between the two tracks, $D_{min}(i,j)$; and the line integral of tubulin intensity in the gap $I_{MT}(i,j)$ (calculation described above). Fluctuation radius, $r_{fluc}$ = 2–3 pixels, was defined as the Brownian diffusion causing fluctuations in plus-end position. For all track candidates $j$ that appear after track $i$ and was considered to be merged with track $i$, the distance between last recorded position of track $i$ and first recorded position of track $j$ [$d_{ends}(i,j)$] was calculated.

Some specific scenarios were considered for a merge and candidates not meeting these were not considered further during optimization. (1) If the frame gap between tracks $i$ and $j$ was zero, $d_{ends}(i,j)$ was required to be within a specified maximum distance, usually similar to $r_{fluc}$, and was set to three pixels in most cases. The occurrence of such gaps was likely in our experimental data and $D_{perp}(i,j)$ was set to $d_{ends}(i,j)$ and $N_{dot}(i,j)$ was assigned to one for these cases. (2) If the tracks $i$ and $j$ are long (>3 x $r_{fluc}$ each), it was ensured that the end point of track $i$ is closest to first position recorded in track $j$. If this condition is not met, the two tracks are not the correct pairs to merge. (3) We calculate

the orientation of track $i$ ($dir_i$), track $j$ ($dir_j$) as well as the orientation of track that would be merged between tracks $i$ and $j$ ($dir_{i-j}$). The candidate track $j$ was rejected if the resulting dot products of orientation vectors are not large enough. Specifically, if the tracks to be merged are small and therefore, their direction estimates carry more error, a maximum angle of 40° is allowed between all pairs: $dir_i$, $dir_j$ and $dir_{i-j}$, otherwise a stricter forward angle of ~20° was enforced for a candidate merge. If these criteria are not met, candidate $j$ was no longer considered for a merge with track $i$. However, these similar criteria cannot be applied to catastrophes because an obtuse angle must be allowed between two tracks flanked by a catastrophe event. (4) For catastrophe cases, a special set of criteria was enforced: tracks $i$ and $j$ should have direction within ~30° of each other. If the distance $d_{ends}(i,j)$ is significantly above $r_{fluc}$, then the to-be-merged region between tracks $i$ and $j$ should also follow a similar angle criteria with tracks $i$ and $j$'s directions and if not, a loose angle criteria of within 90° was enforced.

During gaps, MT plus-ends in our experiments were found to often be in slower growing states, where EB1 comet was not distinctly seen. We refer to these events as pauses. To merge the candidate pauses, $d_{ends}(i,j)$ was ensured to be roughly around $r_{fluc}$, usually set to three pixels while the number of frames that the pause state lasts less than a specified frame length - 15 frames. For these candidate merges, $D_{perp}(i,j)$ and $D_{min}(i,j)$ were set to $d_{ends}(i,j)$, directional overlap $N_{dot}(i,j)$ to 1 (maximum value). Greedy merge of pauses was performed by ensuring $l_{MT}(i,j)$ to be greater than $l_{cutoff}/4$ or $d_{ends}(i,j)$ less than ~1 pixel. If these conditions were met, track $j$ was merged with track $i$. If the gap between tracks $i$ and $j$ did not classify as a pause, two classes of forward gaps were observed in the data: long frame gaps with net displacement limited by (maximum net plus-end speed, set to ~2 pixels per frame) x (number of gap frames, $t_{gap}(i,j)$), or short frame gaps (maximum value of 6 frames allowed) with larger displacement given by (maximum plus-end speed for short frame gap, set to ~5 pixels per frame) x (number of gap frames). Similarly, for catastrophes, a maximum distance condition was imposed (10 pixels). For such candidate merges (forward gap and catastrophes), $D_{perp}(i,j)$ was calculated from cross product between tracks $i$ and $j$ and if found greater than a maximum value of 10 pixels, the candidate merge was rejected. $D_{min}(i,j)$ was set to $d_{ends}(i,j)$. $N_{dot}(i,j)$ was calculated as the dot product of the orientation of track $i$ and the gap between tracks $i$ and $j$ for forward gap, and product of orientation of tracks $i$ and $j$ for catastrophes.

## Greedy optimization

We note that the tracks to be merged were considered in the order of their appearance that is tracks that emerge sooner were assigned first to the possible partners that emerged after, resulting in a temporally greedy assignment. For all candidate tracks $j$ that can be merged with track $i$, reject candidates where $l_{MT}(i,j)<l_{cutoff}$. Cost metric for optimization was defined as [$1.03\hat{\ }t_{gap}(i,j)$ x $D_{min}(i,j)$], and the optimal track $j$ to be merged with track $i$ was found by minimizing this cost and this minimum cost assignment for merged with track $i$.

Finally, the gaps to be merged were closed. MT tracks shorter than six frames were discarded after the process before extracting the relevant parameters described below.

## Parameters from tracking microtubules in branched networks

### Length distribution of microtubules

Length distribution of MTs in a branched network at a specific time point from the emergence of the first MT was calculated using the MATLAB script *lengthMeasurement.m* and accompanying *findArcLength* and *interpolateTracks* functions provided in *Supplementary file 1*. First, for every MT that was tracked, an interpolated MT track were obtained by estimating the position of the plus-end for all the breaks between consecutive EB1 track stretches. The missing plus-end positions were estimated based on average interpolation in the gaps. A smoothing spline was fit to the entire interpolated MT track to decrease the noise from the positions obtained via tracking during length estimation. Start and end positions of each MT track were weighted higher during the spline fit. The length of MTs were then calculated from the fitted spline. MT lengths in the branched networks were pooled from multiple networks and plotted over time (*Figure 1—figure supplement 2D*), while the probability distribution of lengths at a specified time was displayed in *Figure 1—figure supplement 2E*.

## Growth speed distribution and net-speed measurement

Growth speed of the plus-end was obtained from the EB1 tracks before merging process using *uTrack*'s post-tracking module. Growth speed of each EB1 track in a branched network was measured, data was pooled from multiple networks and displayed in the histogram (*Figure 1—figure supplement 2B*). The net speed of plus-end used for simulating the networks was calculated as follows. MT lengths in branched networks were measured as described above. For each MT track that spanned more than 50 frames (or 100 s), the net speed was calculated by dividing the final length of MT by the time it persisted. This measurement was made for 115 MTs across multiple branched networks. The net plus-end speed ($v_{pe}$) was found to be $5.3 \pm 1.4$ µm min$^{-1}$.

## Number of EB1 comets over time

The number of EB1 comets detected in each frame for a branched network was used as a proxy for the number of MTs present and plotted over time (*Figure 1—figure supplement 2C*). Data regression to an exponential curve over time $a \exp(kt)$ was performed using curve fit function in MATLAB, where $k$ is the effective branching nucleation rate.

## Branching angle distribution

For every new nucleation event in a branched network, a mother MT was assigned via greedy optimization using *findParentTrack.m* accompanying *fillLines* provided in **Supplementary file 1**. As before, to find the mother MT $j$ for track $i$, a number of quantities were recorded for all candidates $j$: the direction dot product $N_{dot}(i,j)$; minimum cartesian distance $D_{min}(i,j)$ and perpendicular distance $D_{perp}(i,j)$.

First, for each track, the direction and straight-line fit coefficients were obtained. Starting from the last track $i$ that appeared in the stack, all the tracks that came before it was considered a mother candidate. For each mother candidate track $j$, the position along the track $j$ that is closest to the start of track $i$ was found, $p_j(i)$ and the cartesian distance between this point and start of track $i$ was assigned to $D_{min}(i,j)$. $N_{dot}(i,j)$ was found as dot product between start of track $i$ and track $j$ near position $p_j(i)$. Similarly, $D_{perp}(i,j)$ was found between the two constructed lines – one at the start of track $i$ and other near $p_j(i)$ for track $j$. Similar to track merging, a few specific scenarios were considered. (1) The candidate parent $j$ was rejected if these either the dot product is too low (maximum angle allowed between mother and daughter was 150°) or cartesian and perpendicular distances too high (20 pixels maximum allowed for each). Additionally, if the point $p_j(i)$ was recorded after the start of track $i$, then this candidate cannot be the mother and was therefore, rejected. (2) If $D_{perp}(i,j)$ was found high (>8 pixels) or dot product $N_{dot}(i,j)$ low (<9° angle), the track $i$ was extrapolated to find an intersection with track $j$. The point in track $j$ that is closest to the intersection with found and it was ensured to have recorded before the first point in track $i$. In this case, $D_{min}(i,j)$ was reassigned to the cartesian distance between this intersection point in track $j$ and first position in track $i$. (3) If the branched track $i$ crosses over its parent $j$, then candidate parent track $j$ was rejected.

For track $i$, all candidates $j$ that were not rejected were considered and cost metric $D_{min}(i,j)$ was minimized. If there are a lot of possible parents, the cost metric was set to $[D_{min}(i,j)+1- N_{dot}(i,j)]$. The mother with minimum cost was assigned to track $i$ and the angle between these tracks was calculated as the branching angle.

To pool measurements from various branched networks over time in *Figure 1—figure supplement 2C–E*, time point 0 was considered as the nucleation of the first mother MT (de novo nucleation event) for each network, which accounts for stochasticity in the time of network origin.

## Simulation of the single-step and sequential models

Stochastic kinetic simulations of branched MT networks were generated in MATLAB. For both models, two parameters: net growth speed of MT plus-ends ($v_{pe} = 0.09$ µm sec$^{-1}$) and branching angle distribution were obtained from experiments. Stochastic simulations were started at time 0 s with one MT of zero length (de novo nucleation) and all MTs thereafter were only nucleated by branching from existing ones. MTs were approximated to grow constantly with prescribed net growth speed ($v_{pe}$), and the two simulation models were set up as following.

## Single-step model

For single-step model, only one parameter: the binding rate constant of the nucleators ($k$) was defined (units: $\mu m^{-1}\ sec^{-1}$). The generation of new branched MTs was simulated as a Poisson process as following. For every time step ($\Delta t$), the new nucleation events were determined by generating a Poisson random number with mean rate constant times time interval and the current total length of MTs, $k\ \Delta t\ L(t)$. The position of new nucleation event was defined as a random location along the existing MT length and the angle of branching was selected from gaussian distribution with mean 0° and standard deviation 9°. The plus-end of newly nucleated MT also grows with the net growth speed. These steps were repeated iteratively for a fixed time interval to generate a branched network via the single-step model. The simulation code is provided in the file *singlestepModel.m* in *Supplementary file 1*.

One model parameter $k$ in the single-step model was set as follows. The length of mother MT before the first branching nucleation event occurred was measured experimentally. $k$ in single-step model was set to $1.1 \times 10^{-3}\ \mu m^{-1}\ sec^{-1}$ such that the mean mother MT length matched the experimental data.

## Sequential model

For the sequential model, two parameters were required: the binding rate constant for deposition of nucleation sites ($k_{bind}$, units: molecules $\mu m^{-1}\ sec^{-1}$) and branching rate constant ($k_{branch}$, units: $sec^{-1}$ molecule$^{-1}$). For generating a new branched MT, the first step was to deposit nucleation sites to the existing MT mass and this step was simulated as a Poisson process as following. For every time step ($\Delta t$), the new binding events were determined by generating a Poisson random number with mean rate constant times time interval and the current total length of MTs, $k_{bind}\ \Delta t\ L(t)$. The position of deposited site was defined randomly along the existing MT length. Branched MTs were then nucleated from these bound, inactive sites with another Poissonian step: for every site at every time step ($\Delta t$), whether a branch occurs was determined by generating a Poisson random number with mean branching rate constant times time interval, $k_{branch}\ \Delta t$. We then generate a zero-length MT, with branch angle was selected from a gaussian distribution with mean 0° and standard deviation 9°, and grows over time with the pre-defined plus-end growth speed. These steps were repeated iteratively to generate a branched network via the sequential model. The simulation code is provided in the file *sequentialModel.m* in *Supplementary file 1*.

Two model parameters $k_{bind}$ and $k_{branch}$ in the sequential model was set as follows. As in the single-step model, the length of mother MT when the first branching nucleation event occurred was measured and compared between experimental and simulated branched networks. This sets the net rate of nucleation, approximately proportional to $k_{bind}\ k_{branch}$ at initial times, in the model. The individual parameters were set by characterizing how the spatial bias (*Figure 2—figure supplement 1C*) varied with individual values of $k_{bind}$ and $k_{branch}$. The dimensionless ratio of the activation rate constant, $k_{branch}$, and effective binding rate constant, $\sqrt{(k_{bind}\ v_{pe})}$ were varied and the rescaled bias (*Figure 2—figure supplement 1C*) was plotted. The derivation of this dimensionless number is shown in Appendix 1. Individual parameter values were chosen where the branching rate was slower than the binding of sites and robust bias was produced for the first branching nucleation event along the mother MT. Final parameters used: $k_{bind}$ = 0.1 molecules $\mu m^{-1}\ sec^{-1}$, and $k_{branch}$ = 2.5×10$^{-4}$ sec$^{-1}$ per molecule.

For all data reported in this study, 250–4000 independent of simulations of branched network from the single-step model or the sequential model were performed, all data was pooled and reported.

## Measurement of first branching nucleation site

As a readout for branching architecture, the position of first branching nucleation event along the naked mother MT (*de novo* nucleation) was measured experimentally (*Figure 2* and *Figure 2—figure supplement 1*) and compared with simulated models. Time-lapse movies of experimental branched MT network were recorded as described above. For each observed branched network, the location of minus-end of first (*de novo*) nucleated MT was recorded ($m_0$). The position of minus-end of the first mother does not change over time because the transport of MTs in this assay with sodium orthovanadate. At time point where the first branched MT was nucleated along the mother ($t_1$), the

position of the minus-end of the new nucleation ($m_1$) as well the plus-end of the mother MT ($p_0$) were recorded. The distance of nucleation site from the mother's minus-end ($|m_0-m_1|$, *Figure 2—figure supplement 1B*), and the mother's plus-end ($|p_0-m_1|$, *Figure 2A*), as well as fractional distance along the mother filament $\left(\frac{|m_0-m_1|}{|m_0-p_0|}, \text{Fig. 2B}\right)$ were measured. These measurements were made for both experimental and simulated branched networks and their probability distribution were plotted (*Figure 2A–B*). These positions were logged manually for experimental networks to ensure accuracy of the measured nucleation profile. For experimental data, n = 381 independent branched networks were observed with at least three independent extract preparations. n = 4000 simulations were made for single-step model and sequential model each. All data was pooled for both experiments and simulations and compared. 95% confidence interval on probability distributions was obtained by bootstrapping.

## Distribution of minus-ends and plus-ends in dense branched networks

MT plus-end distribution was measured in the experimental branched network as follows. EB1 comets in branched networks were imaged as described above. Image stacks with growing branched networks were cropped and EB1 comets were detected with the procedure described above. Moving median filter of 20 frames was applied and comet detection parameters input to *uTrack* were set so that false positive EB1 detections were minimized. Specifically, threshold of 12 s.d. and step size of 2 s.d. was chosen and all comets in the time sequence were detected. For each branched network, the origin point was manually found and recorded. The cartesian distances of all comets were measured from the origin and rescaled by the longest distance, a proxy for length of branched network, at each time point. Data from multiple networks generated was pooled for a specified number of comets in each network (between 10–55 comets for *Figure 2D*) and compared to simulated networks.

For simulated networks obtained from the single-step and sequential model, analogous measurements were made. The cartesian distance of all MT plus-ends from the origin of the network was calculated and rescaled by the longest distance (length of branched network). Similarly, the cartesian distances of all MT minus-ends from the network origin were calculated and normalized by the length of branched network. Data from 250 simulated networks for each model was pooled for a specified number of MTs (between 10–60 MTs reported in *Figure 2D*) and compared with experimental measurements.

## Distribution of tubulin intensity in dense branched networks

To obtain the tubulin intensity distribution in experimental branched networks, snapshots of a large number of branched networks were obtained at a specified time point. Here, time-lapse of networks was avoided so that the tubulin signal on pre-existing MTs does not bleach over the course of recording. The script for measuring tubulin intensity in experimental networks is *tubulinIntensity.m* in *Supplementary file 1*. First, the background illumination of the MT channel was corrected for the entire field of view using *illumCorrect.m* in *Supplementary file 1*. The image was filtered with a Gaussian filter of size 110 pixels and sigma 30 pixels. Additionally, a blank image with no MTs in the field of view was recorded with identical imaging settings and also filtered with same Gaussian filter. Because in our imaging setup, uneven illumination is caused by the intensity decay of the gaussian beam near the edges of the image acquired with a large sCMOS chip, intensity was homogenized by dividing the filtered image with the filtered blank image. Individual branched networks in the field of view were isolated after intensity correction, their boundary was manually drawn in ImageJ, the coordinates were saved and read using MATLAB scripts. First, a convex hull was generated in MATLAB after using Delaunay triangulation and intensity of each pixel inside the hull was recorded alongside the cartesian distance of that pixel from the branch network origin. Background tubulin intensity was calculated from the mean intensity of all pixels in the image that lie outside of the convex hull, subtracted from all pixel intensities inside the hull, which were then normalized by the mean intensity inside the hull. Normalized tubulin intensity in the branched network versus distance was then plotted and compared with simulated networks (*Figure 2E*).

To obtain an analogous measurement of tubulin intensity in simulated networks, an equivalent 'image' of simulated network was generated at a specified time point and blurred with a gaussian function of size 30 pixel and sigma 1.5 pixels, representing the point spread function of our

microscope. Convex hull on the MT signal was produced exactly as described above. Tubulin intensity inside the hull was calculated and the corresponding cartesian distance from the network origin recorded. The tubulin intensities were then normalized by their mean and compared to the experimental results (*Figure 2E*).

## Acknowledgements

We thank Dr. Christiane Wiese for generously providing XenC antibodies. AT thanks the Petry lab members particularly Ray Alfaro-Aco for discussions, Jae-Geun Song and Brian Mahon for purification of augmin, Ben Bratton for advice on image analysis, CIAN, OMIBS and Physiology courses at MBL for training. This work was supported by the American Heart Association predoctoral fellowship 17PRE33660328 (to AT), the NIH New Innovator Award 1DP2GM123493, Pew Scholars Program in the Biomedical Sciences 00027340, David and Lucile Packard Foundation 2014–40376 (all to SP), and the Center for the Physics of Biological Function sponsored by the National Science Foundation grant PHY-1734030.

## Additional information

### Funding

| Funder | Grant reference number | Author |
|---|---|---|
| American Heart Association | 17PRE33660328 | Akanksha Thawani |
| National Institute of General Medical Sciences | 1DP2GM123493-01 | Sabine Petry |
| Pew Charitable Trusts | 00027340 | Sabine Petry |
| David and Lucile Packard Foundation | 2014-40376 | Sabine Petry |
| National Science Foundation | PHY-1734030 | Joshua W Shaevitz |

The funders had no role in study design, data collection and interpretation, or the decision to submit the work for publication.

### Author contributions

Akanksha Thawani, Conceptualization, Software, Investigation, Methodology, Funding acquisition, Writing—original draft, Writing—review and editing; Howard A Stone, Conceptualization, Supervision, Writing—review and editing; Joshua W Shaevitz, Sabine Petry, Conceptualization, Supervision, Funding acquisition, Writing—review and editing

### Author ORCIDs

Akanksha Thawani https://orcid.org/0000-0003-4168-128X
Howard A Stone https://orcid.org/0000-0002-9670-0639
Joshua W Shaevitz https://orcid.org/0000-0001-8809-4723
Sabine Petry https://orcid.org/0000-0002-8537-9763

### Ethics

Animal experimentation: This study was performed in strict accordance with the recommendations in the Guide for the Care and Use of Laboratory Animals of the National Institutes of Health. All of the animals were handled according to approved Institutional Animal Care and Use Committee (IACUC) protocol # 1941-16 of Princeton University.

### Decision letter and Author response

Decision letter https://doi.org/10.7554/eLife.43890.027
Author response https://doi.org/10.7554/eLife.43890.028

## Additional files

### Supplementary files

• Supplementary file 1. MATLAB-based image analysis and simulation software.
DOI: https://doi.org/10.7554/eLife.43890.023
• Transparent reporting form
DOI: https://doi.org/10.7554/eLife.43890.024

### Data availability

All data generated or analyzed during this study are included in the manuscript and supporting files.

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

# Appendix 1

DOI: https://doi.org/10.7554/eLife.43890.025

## Analytical equation for the sequential reaction model

We consider a simplified model for the sequential reaction scheme. Here, the first reaction step deposits $S(t)$ sites on the existing MT lattice in time $t$ seconds. The second reaction step subsequently generates branch MTs $N(t)$ from the deposited sites. The deposition of nucleation sites occurs on the existing MTs of average length $L(t) \approx v_{pe}t$, where $v_{pe}$ is the net growth rate of the plus-end.

The rate of site deposition is expressed as,

$$\frac{dS(t)}{dt} = k_{bind}L(t)N(t) \approx k_{bind}v_{pe}tN(t) \tag{1}$$

The rate of nucleation from deposited sites reads,

$$\frac{dN(t)}{dt} = k_{branch}(S(t) - N(t)) \tag{2}$$

Differentiating **Equation (2)** and substituting in (1) gives,

$$\frac{d^2N(t)}{dt^2} + k_{branch}\frac{dN(t)}{dt} = k_{branch}k_{bind}\left(v_{pe}t\right)N(t) \tag{3}$$

Non-dimensionalizing equation (3) with dimensionless time, $\tau = k_{branch}t$,

$$N_{\tau\tau} + N_{\tau} = \frac{k_{bind}v_{pe}}{k_{branch}^2}\,\tau N \tag{4}$$

The dimensionless ratio of the branching rate constant to the effective binding rate constant $\left(\frac{k_{branch}}{\sqrt{k_{bind}\,v_{pe}}}\right)$ in **Equation (4)** determines the bias in nucleation from older MT lattice near the minus-end as represented in **Figure 2—figure supplement 1C**.

