## [Decision Letter]

[Editors’ note: this article was originally rejected after discussions between the reviewers, but the authors were invited to resubmit after an appeal against the decision.]

Thank you for submitting your work entitled "Spatiotemporal organization of microtubules in branched networks" for consideration by *eLife*. Your article has been reviewed by two peer reviewers, and the evaluation has been overseen by a Reviewing Editor and a Senior Editor. The reviewers have opted to remain anonymous.

The reviewers agree that the image analysis and other technical aspects of the assay are significant, but they have both raised substantive questions about the overall significance of the results. This includes aspects of the model analysis and also of the experiments. Please see below for the full reviews.

Taken together, in our joint discussions of the paper, we have concluded that in its present form your paper does not represent a substantial enough advance to be published in *eLife*. We would in principle be prepared to consider a new submission if the concerns of the reviewers are fully addressed. The paper and the point-by-point rebuttal would then most likely be sent to the same reviewers.

*Reviewer #1:*

The authors measure the spatial distribution of augmin-mediated 'microtubule-dependent microtubule nucleation sites' in *Xenopus* egg extract using a previously developed assay. They find a bias of nucleation sites being positioned closer to microtubule minus ends and explain this bias using a 2-step kinetic model assuming delayed activation of nucleation sites after their deposition. The quantitative image analysis of the data is novel and seems quite advanced allowing the authors to quantitatively compare their data with their preferred model and an alternative simpler 1-step kinetic model. To provide additional experimental support for the preferred model the authors change the concentration of augmin (partial depletion) and of TPX2 (partial depletion and addition) in an attempt to influence the first and second kinetic step, respectively, of their 2-step model. Overall, this study is technically interesting and could provide insight into the mechanism of branched microtubule nucleation in *Xenopus* egg extract if predictions of the model could be tested more directly.

Main concerns:

1) Validation of the model: The model which has more free parameters produces a better fit to the data. How strong is the evidence that this model uniquely describes the data best? For example, one may wonder if a cooperative (autocatalytic) 1-step binding model could also explain the experimental data. The two models examined here make different predictions for the kinetics of binding and the resulting distributions of the 'nucleator' and, in the 2-step model, of the 'activator'. The rationale of the experiments in Figure 4 is based on assuming that augmin is the 'nucleator' and TPX2 is the 'activator'. However, that is not demonstrated. The distributions of these molecules along microtubules using fluorescent constructs could easily be measured and compared to the model predictions. Ideally also the predicted rather slow binding kinetics could be measured. These experiments could lend solid experimental support for the assumptions made about the roles of these two molecules in the context of the kinetic model, and hence validate the model. Otherwise the study remains rather technical and speculative.

2) Significance and agreement with literature: as the authors discuss, in spindles, particularly in large *Xenopus* egg extract spindles, there is a RanGTP gradient which should result in the highest TPX2 activities being present in the spindle center which could remove the nucleation bias towards minus ends that the authors observe in their assay where there is no RanGTP gradient. A bias of nucleation toward spindle poles seems to contradict the findings of at least some previous literature, e.g. Brugues et al., 2012, that concludes that most nucleation takes place in the spindle center. Therefore, the question arises how much the particular assay conditions (uniform distributions of constitutively RanQ69L and the non-specific ATPase inhibitor vanadate) reflect the distribution of nucleation sites in spindles. In other words, why does the study advance our understanding of spindle assembly?

3) Technical question: how can the authors know that their image analysis procedure correctly connects several EB1 tracks and assigns branching points to the correct microtubule in what seems to be a crowded microtubule bundle imaged with standard light microscopy resolution?

*Reviewer #3:*

This manuscript the role of Augmin-mediated microtubule nucleation of the side of existing microtubules in vitro using a *Xenopus* extract system to determine what factors control the shape of the branched network that forms. The authors have worked hard to rigorously quantify the structure and dynamics of the networks that form in their system and have compared them to simulations, in which the probability of branching depends solely on the binding of the Augmin/ γ-TuRC complex or also requires a second activation step. Their data best fit the simulations with an activation step, leading them to conclude that branching is a two-step process that requires binding of Augmin/γ-TuRC followed by activation by TPX-2.

The authors have developed a very nice assay for the growth of branched microtubule networks, but I was disappointed that they did not do more with it, for example by testing different models in their simulations or by investigating the nature of the activation step in microtubule nucleation. As it is, they describe a very nice assay and analysis pipeline but their results represent only a small step forward. I am not sure whether this is enough for *eLife*, given that the Petry group have already shown that TPX2 is an activator of the Augmin/γ-TuRC.

1) They describe their results as explaining why branching is biased towards the minus ends of the microtubules, which is correct, but misleading. The probability of branching is proportional to the age of each region of the microtubule, which is related to proximity to the minus ends, as these are the oldest regions. Nevertheless, the pattern would be different if the microtubules were growing more slowly.

2) The experiments in Figure 4 are important, but seem incomplete as they do not clearly distinguish between the roles of Augmin (part of the nucleator) and TPX2 (the presumed activator).

3) Why was the *k_act_* chosen to be 0.07 in the simulation in Figure 4B rather than 0.2? This has the effect of flattening the orange line in the simulation to more closely resemble the results from TPX2 depletion in Figure 4C, but this is not a convincing demonstration of anything as reducing the *k_bind_*to 0.07 might have a very similar effect.

4) The simulations use a *k_binding_* for the nucleator complex, but do not include a *k_off_*, implying that the nucleator binds irreversibly to the side of the microtubule. Having the nucleator also disassociate at a certain rate does not make much difference to the nucleator only model, where the branching microtubule starts growing immediately, but will affect the pattern of branching if the nucleator needs a second activation step, as this will reduce the bias towards the old sections of the microtubule.

---

## [Author Response]

[Editors’ note: the author responses to the first round of peer review follow.]

Reviewer #1:Main concerns:1) Validation of the model: The model which has more free parameters produces a better fit to the data. How strong is the evidence that this model uniquely describes the data best? For example, one may wonder if a cooperative (autocatalytic) 1-step binding model could also explain the experimental data.

We thank the reviewer for this important question. The development of our model was partly data-driven. We found that branched MTs originated from the older MT lattice near the minus-end of the mother MTs, even though there is a lot of MT lattice available near the plus-ends where the effectors could bind to generate a new branch. This is most clearly evident with the site of the first branching event from a naked mother MT (Figure 2B). Hence, we proposed that branching sites are being “marked”, due to more binding events to older MT regions, and a new branch is nucleated with another Poisson process. Thus, any 1-step model where there is no time delay between the deposition of nucleation site and the generation of daughter, even if autocatalytic, will always result in a flat probability distribution for Figure 2B. Hence, the single-step model, which we define as binding and branching being concurrent, does not capture our measured distributions. In contrast, the sequential model, consisting of nucleation site deposition followed by branch formation, describes the data best, as depicted in Figures 1 and 2. This is not simply because of more available parameters, but rather a consequence of the molecular steps involved. In fact, the two models are related: when the second rate of branch formation is sped up, the sequential model essentially becomes again the 1-step model that no longer captures the branching network architecture (Figure 2—figure supplement 1C). To ensure that this is also understood by a wide audience, we incorporated this link between the models in the subsection “Measuring the first branching events in branched microtubule networks”, and made an effort to better explain the models in the revised Results sections.

The two models examined here make different predictions for the kinetics of binding and the resulting distributions of the 'nucleator' and, in the 2-step model, of the 'activator'. The rationale of the experiments in Figure 4 is based on assuming that augmin is the 'nucleator' and TPX2 is the 'activator'. However, that is not demonstrated. The distributions of these molecules along microtubules using fluorescent constructs could easily be measured and compared to the model predictions. Ideally also the predicted rather slow binding kinetics could be measured. These experiments could lend solid experimental support for the assumptions made about the roles of these two molecules in the context of the kinetic model, and hence validate the model. Otherwise the study remains rather technical and speculative.

We are grateful for the reviewer’s insightful suggestions as they prompted us to invent new ways to investigate this question. In our revised manuscript, we address this comment with the sequential experiments and direct measurement of binding rate of TPX2 to microtubules, as described in detail in the common response above. In the process, we made measurements such as direct, single-molecule observations of microtubule-associated proteins in *Xenopus* egg extracts, which we believed not to be possible. This has now opened a way of accessing the kinetics of proteins in their cytoplasmic environment, which had not been accessible previously. We believe this will be an important breakthrough for our and many other labs that work between in vivo and in vitro MT systems. Most importantly, these experiments resulted in a whole new figure (Figure 4), which we believe is now the highlight of this manuscript. The big surprise is that these experiments established a molecular order which could not have been predicted: our new results altogether show that branching is a sequential pathway that involves TPX2’s binding to individual MTs followed by the presence of augmin/γ-TuRC prior to branch nucleation.

2) Significance and agreement with literature: as the authors discuss, in spindles, particularly in large *Xenopus* egg extract spindles, there is a RanGTP gradient which should result in the highest TPX2 activities being present in the spindle center which could remove the nucleation bias towards minus ends that the authors observe in their assay where there is no RanGTP gradient. A bias of nucleation toward spindle poles seems to contradict the findings of at least some previous literature, e.g. Brugues et al., 2012, that concludes that most nucleation takes place in the spindle center. Therefore, the question arises how much the particular assay conditions (uniform distributions of constitutively RanQ69L and the non-specific ATPase inhibitor vanadate) reflect the distribution of nucleation sites in spindles.

These points by the reviewer are excellent, which we address below in multiple paragraphs including clarification and reasoning behind our assay setup.

a) Clarity: Based on the reviewer’s excellent points above, we realized that our description of the results needed improvement. First, we note that the described minus-end bias is in essence due to the longer lifetime of those regions of the MT lattice compared to the newly formed lattice near the plus-ends. This was also pointed out by reviewer 2 in their comments and thereby addressed. In our sequential model, new region of lattice near the plus-end is formed while the older lattice near the minus-end collects larger number of molecules of branching factor 1 (TPX2), making it more likely to generate a branch from the subsequent presence of augmin/γ-TuRC. In multiple places throughout the manuscript, we have clarified this point by stating, “older lattice near the minus-ends”.

b) RanGTP: We agree with the reviewer that the spindle assembly factor TPX2 will be released near chromosomes. However, the length scale and profile of RanGTP gradient have been shown to not bear any importance for the spindle structure (Oh et al., 2016). Here, the authors showed that the MT binding of spindle assembly factors such as TPX2, not their concentration profile, drives spindle assembly. Therefore, the homogeneous presence of RanGTP as in our assay still provides useful information on how the spindle is organized as summarized below. In fact, we believe that not having a spatial cue (e.g. a Ran gradient) was necessary for us to study how molecular effectors regulate MT nucleation in space and time and organize the MT networks, which is a first step towards obtaining a more advanced understanding of how the spindle is assembled.

c) Molecular motors: To address how underlying molecules result in self-organized structures such as the branched networks, it was necessary to decouple nucleation of new MTs from their sorting by molecular motors, as detailed below. In this work, we used the sodium orthovanadate to inhibit all motor activity, instead of using a cocktail of drugs to inhibit the dominant motor activities such as Eg5 and dynein, which had been done in previous works (Brugues et al., 2012). In our previous work (Supplementary Figure 1 in Petry et al., 2013), we showed that the concentration of sodium orthovanadate used resulted in structures that look identical to when only dynein is inhibited with p150-CC1. Inhibition of active transport in this work was necessary to make accurate measurements so that net translation of microtubules does not induce systematic errors in our measurements of nucleation. In the case of net translation of microtubules, the tracking of microtubules by following the motion of EB1 comets at the plus-ends does not provide us the information on growth of microtubules, which was necessary for our analysis. Therefore, to study the nucleation and growth of microtubules in branched networks, it was important to ensure that motor-based microtubule transport was abolished and we used sodium orthovanadate to achieve this.

Additionally, we note that our measurements recapitulate the measurements that have been made for microtubules in the spindle, such as the growth speed, predictions of nucleation rate, and microtubule length distribution. Therefore, the branched networks reflect the properties of the entire spindle.

A bias of nucleation toward spindle poles seems to contradict the findings of at least some previous literature, e.g. Brugues et al., 2012, that concludes that most nucleation takes place in the spindle center.

We thank the reviewer for highlighting this. As described in paragraph (a) above, our measurements reflect bias in nucleation from older MT regions near the minus-ends, and not from spindle poles per se. Notably, many excellent studies including ones from Needleman and Brugues labs have shown that minus-ends (or older lattices) do not just exist near the spindle poles, but throughout the spindle. Based on our work, older lattice regions near these minus-ends near the spindle center could serve as preferential sites for branching nucleation. Additionally, our mechanism could result in specific amplification of MTs that stably attached to kinetochores (longest-lived MTs) near the spindle center and hence generate denser k-fibers. We now include this in the Discussion “[…] the preferential nucleation from older MT lattice regions could allow for precise control over where branching occurs in the spindle. We propose that TPX2, which is released in the vicinity of chromosomes (Kalab et al., 2002; Oh et al., 2016), accumulates on the older MT lattice regions near chromosomes such as on MTs captured and stabilized by the kinetochores (Cimini et al., 2003; DeLuca et al., 2011; Lampson et al., 2004; Umbreit et al., 2012) and older lattice regions such as minus-ends near the spindle center. Branching MT nucleation by augmin/γ-TuRC from deposited TPX2 sites will selectively amplify older MTs generating dense kinetochore fibers near chromosomes.”

Thus, our work is congruent with previous studies that suggested MT nucleation, potentially by branching, mostly taking place in the spindle center. Further sorting of these MTs by motor activity may generate the spindle architecture, as we now discuss in the last paragraph of the Discussion. More generally, we note that how spatial and temporal hierarchy the MT nucleation pathways are related to one another, and how MT transport fuses these pathways to create a uniform MT nucleation profile that maintains the spindle, is the next big open question in the field.

In summary, based on our results and previous literature, we hypothesize that even with the presence of a Ran gradient and motor activity, the preference of MT nucleation from the older lattice will still persist. As such, we do not think that our results contradict the previous work, and rather advance the understanding of how MT generation is regulated within the spindle. We now discuss this comprehensively in our revised Discussion, fourth paragraph (Ran gradient) and last paragraph (motor activity).

In other words, why does the study advance our understanding of spindle assembly?

Impact: We note that the few existing studies on nucleation within the spindle measure profiles at a single snapshot during metaphase when the entire spindle has been assembled and a lot of microtubule mass already exists and has been sorted by the motors. Here, the MT nucleation pathways are no longer distinguishable and intersect, driven by molecular motors, to yield the uniform MT nucleation profile that characterizes the spindle. Therefore we have a limited understanding of how a single MT nucleation pathway is formed and regulated.

Our work describes such an individual MT nucleation pathway for the first time, which corresponds to the dynamic process of generating the microtubule mass within the spindle and is therefore is essential for how the spindle is made. Our study demonstrates how the age of the MT lattice has a more significant role than previously anticipated for autocatalytic nucleation as a result of our experimentally demonstrated sequential kinetic scheme, and therefore advances our understanding of how MT nucleation within the spindle supports spindle assembly.

Most importantly, with the experiments performed during the review process, our study now establishes how the kinetics and molecular sequence of the branching effectors result in the geometry and network-wide architecture of branched networks. This had been a major missing area in this field, with very few previous examples that have only recently emerged (Roostalu et al., Cell 2019), and will therefore advance the spindle assembly field. Additionally, we believe that the new insights about the branching mechanism produced in this work informs bottom-up in vitro reconstitution of this pathway.

In light of this comment by the reviewer, we have thoroughly edited and expanded our discussion of this subject in the manuscript to cover three points: the mechanism of branching (Discussion, second paragraph), comparison of our measurements to previous work (Discussion, third paragraph), implications for spindle assembly (Discussion, fourth paragraph). We hope that our manuscript now addresses this point and is more impactful for the broad readership.

3) Technical question: how can the authors know that their image analysis procedure correctly connects several EB1 tracks and assigns branching points to the correct microtubule in what seems to be a crowded microtubule bundle imaged with standard light microscopy resolution?

We thank the reviewer for this question. We ensured that the analysis procedure was reliable as follows. First, we imaged branched networks from their initiation at a high frame rate imaging of 0.5 frames per second, while imaging both the EB1 and tubulin channels at every frame and limited our analysis to branched networks at early time points when the density of microtubules is not very high. Our analysis pipeline was benchmarked by manually tracking a few branched networks and comparing it with the analysis. Most importantly, for all measurements displayed in our work using the analysis pipeline (Figure 1; Figure 1—figure supplements 1 and 2), the analysis was conducted only for branched networks which had on average 20 microtubules. Therefore, we stopped our analysis before the network became too crowded, and analyzing multiple networks at low microtubule density and high frame rate was sufficient to provide the parameters we needed. For all networks tracked and used to obtain measurements, we visually inspected for inaccuracies in all networks we analyzed with the pipeline, and found that the analysis scheme had robust performance for low density of tens of microtubules per network. With this, we measured the reliable, bulk parameters that were then used to set up the model simulations – growth speed and branch angles.

Lastly, to conclusively accept or reject our kinetic models (new Figure 2), only the most reliable experimental measurements were used, where there was no ambiguity in locating the branch point manually. Specifically, Figure 2A-B report the location of the very first branching event on a previously unbranched (de novo) mother microtubule. See Figure 2—figure supplement 1 for two representative examples. Because crowding by other microtubules is not a concern for this measurement, dual-channel time lapses at high frame rate were sufficient to ensure accuracy. For the plots in Figure 2C-E at high microtubule density, branch points were not extracted, hence the minus-ends plot in Figure 2C does not show an experimental curve. For large networks, we measured the profile of plus-ends (Figure 2D), which can be unambiguously obtained by isolating EB1 particles, or tubulin intensity profile (Figure 2E), which was again measured over background; both are described in Materials and methods and illustrated in Figure 2—figure supplement 2.

Reviewer #3:[…] The authors have developed a very nice assay for the growth of branched microtubule networks, but I was disappointed that they did not do more with it, for example by testing different models in their simulations or by investigating the nature of the activation step in microtubule nucleation. As it is, they describe a very nice assay and analysis pipeline but their results represent only a small step forward. I am not sure whether this is enough for eLife, given that the Petry group have already shown that TPX2 is an activator of the Augmin/γ-TuRC.

We thank the reviewer for seeing the potential in our work and for suggesting avenues for improvement, which prompted us to further investigate our model and the mechanism we propose. Our revised work now comprehensively establishes how the kinetics and molecular sequence of branching effectors generate the architecture of branched networks, and additionally provides insights into the role of augmin, TPX2 and γ-TuRC in the branching pathway. As such, we believe that the revised work is of higher significance to the mitotic spindle and cytoskeleton fields.

We note that the focus of our work is to understand how the reaction kinetics of branching effectors establishes branched networks. Explaining how the regulation of MT nucleation can lead to specific architectures is also relevant for other MT nucleation pathways and the cytoskeleton as a whole.

Because our readouts are limited to measurement of kinetics, distinguishing between the molecular mechanisms of activation is not a possibility with this work. To investigate the molecular mechanism of branched formation or the activation as the reviewer suggested, a reconstituted system built from purified components is needed, and required live studies via single molecule microscopy, as well as atomic structures of these molecules in various combinations and in various states. At this point, we also note that we had previously identified that TPX2 had specific sequence signatures that suggested similarity with the γ-TuRC activator sequence (Alfaro-Aco et al., 2017). We proposed TPX2’s role in γ-TuRC activation based on mutations in those sequences disrupting the branching activity of TPX2. This proposition does not establish TPX2 as a direct activator of γ-TuRC, and what the function of TPX2 is in the branching pathway is unknown as of yet, as we describe in the last paragraph of our Introduction. Our new experiments (Figure 4) provide insight into the role of TPX2 in branching, but more work is needed to fully understand this including an in vitro reconstitution of branching and directed mutations in TPX2 that disrupt specific functions. While we are interested in pursuing this in the future, such a study is beyond the scope of our current work and will require pursuit by many labs throughout the next decade.

1) They describe their results as explaining why branching is biased towards the minus ends of the microtubules, which is correct, but misleading. The probability of branching is proportional to the age of each region of the microtubule, which is related to proximity to the minus ends, as these are the oldest regions.

We thank the reviewer for this comment. As the reviewer correctly pointed out, our model and new accompanying experiments show preference due to age of the lattice. In our architecture measurements, the minus-ends served as the direct reference and a way of normalizing our nucleation profiles, therefore, we used that as the metric for our measurements. In multiple places throughout the manuscript, we have now clarified the reviewer’s point by stating, “older lattice near the minus-ends”.

Nevertheless, the pattern would be different if the microtubules were growing more slowly.

We note that our model results in two classes of patterns: one where the bias towards the older lattice (minus-ends) is distinct and predicts the pattern we observe in experiments (Figure 2—figure supplement 1C, blue, green, orange and red curves), and the other extreme where branching rate is fast compared to the binding (magenta and purple curves, Figure 2—figure supplement 1C). The latter regime effectively is the single-step model. Further, our sequential model is robust such that changing the parameters by a few fold does not change the effective pattern (Figure 2—figure supplement 1C).

Therefore, decreasing the plus-end growth speed by a few fold does not change the regime of our model to result in a different pattern: the length scale of the pattern will expand proportionally to the absolute growth speed, but the shape of the pattern, especially our rescaled measurements (Figure 2B, C-E), will remain the same. Furthermore, we directly measured the growth speed of microtubules in our *Xenopus* egg extract system (Figure 1—figure supplement 2B). We used these direct measurements as model parameters, and making the MTs grow slower did not correctly represent our experimental system that we aimed to describe in our model.

We have highlighted this point in the manuscript. In the subsection “Measuring the first branching events in branched microtubule networks”, we describe these results and end with, “Thus, our model results in two regimes of network architectures and is not sensitive to the chosen parameters.”

2) The experiments in Figure 4 are important, but seem incomplete as they do not clearly distinguish between the roles of Augmin (part of the nucleator) and TPX2 (the presumed activator).

We thank the reviewer for raising this important point. In the revised manuscript, we performed new experiments summarized in the common response above and described in the new Figure 4. These sequential experiments in conjunction with direct visualization of TPX2, augmin/γ-TuRC establish the molecular sequence of proteins during branching. In summary, our new experiments altogether prove the sequential model where TPX2’s binding to de novo MTs deposits nucleation sites and branches emerge due to available augmin/γ-TuRC complexes.

3) Why was the k_act_ chosen to be 0.07 in the simulation in Figure 4B rather than 0.2? This has the effect of flattening the orange line in the simulation to more closely resemble the results from TPX2 depletion in Figure 4C, but this is not a convincing demonstration of anything as reducing the k_bind_ to 0.07 might have a very similar effect.

We thank the reviewer for this question. The revised Figure 3 (previously Figure 4) now quantitatively compares the effect of changing TPX2 and augmin concentration to an equivalent change in *k_bind_* or *k_branch_*. We use this data to conclude that the concentrations of augmin and TPX2 determine the net rate of nucleation, and therefore their unimolecular binding rates are involved in the branching pathway. The revised Figure 4 now addresses the molecular pathway through sequence experiments and single molecule measurements.

4) The simulations use a k_binding_ for the nucleator complex, but do not include a k_off_, implying that the nucleator binds irreversibly to the side of the microtubule. Having the nucleator also disassociate at a certain rate does not make much difference to the nucleator only model, where the branching microtubule starts growing immediately, but will affect the pattern of branching if the nucleator needs a second activation step, as this will reduce the bias towards the old sections of the microtubule.

This is an excellent point raised by the reviewer. We agree that our current model assumes a net rate of binding of the first protein that deposits the branching sites and does not include unbinding kinetics. This choice was made in our work to minimize the number of parameters that had to be fit when the identity of the molecules or their kinetics was unknown in the previous version. With the new experiments, we have directly measured TPX2’s binding to the branched networks. We find the net rate of deposition of TPX2 of 0.4 molecules/μm/s, which corresponds to the irreversible binding rate we assume in the model.

At the same time, the dissociation of the first branching effector at high rates will decrease the effective bias as the reviewer suggested. Based on this, we performed simulations that include the unbinding of branching effector 1 in our sequential model in the following manner. We kept the branching rate constant for the second step, which was chosen from our model where no unbinding of the first protein occurs (*k_branch_* = 2.5x10^-4^ s^-1^), and added an unbinding rate for the molecules of branching effector 1. As before, we matched the average length of mother microtubule when the first branching event occurred in our model to experimental data (Figure 1—figure supplement 1). In order to do that, when a higher unbinding rate was assumed, a higher binding rate of effector 1 was also necessary. Therefore, we explored the parameter space while satisfying our model constraints and present 3 parameter sets where *k_off_*was included. Parameter set 1 represents the irreversible binding with *k_bind_* = 0.1 molecule μm^-1^ s^-1^ (black curve). Parameter set 2 shows the model with *k_off_*= 0.01 s^-1^ with *k_bind_* = 0.14 μm^-1^ s^-1^ (green curve). Parameter set 3 shows the model with *k_off_*= 0.023 s^-1^ with *k_bind_* = 0.2 μm^-1^ s^-1^ (blue curve) and set 4 shows the model with highest *k_off_*= 0.07 s^-1^ with *k_bind_* = 0.4 μm^-1^ s^-1^ (red curve). In agreement with the reviewer’s comment (see Figure A), the spatial bias is reduced for higher *k_off_*. We first attempted to estimate *k_off_*for TPX2 in branched microtubules from *Xenopus* egg extracts to understand whether the unbinding of TPX2 is important to consider in our model. We spiked 2.5nM GFP-TPX2 in the presence of unlabelled, endogenous TPX2 in *Xenopus* egg extracts and observed the association of single TPX2 molecules to the microtubules in branched networks. A few representative kymographs are displayed in Figure A. Visual inspection of this data already shows that TPX2 has long residence times on the microtubule lattice with some traces extending upto 300 seconds or longer. These long residence times prohibit precise measurements of *k_off_*because the photobleaching timescale under these conditions is between 60100 seconds. Nevertheless, we measured the timescale for molecules that “dissociated” clearly during the imaged timelapse. n=34 measurements are plotted as a histogram. This value results in *k_off_*= 0.01 s^-1^, which is likely an overestimate of the unbinding rate because of photobleaching rate being very similar. We note that low *k_off_*for TPX2 is in agreement with previous reports in vitro (Supplementary Figure 3 in Roostalu et al., 2015). Comparing this to our simulations, *k_off_*= 0.01 s^-1^ gives similar degree of bias to when irreversible binding was assumed, which makes us conclude that the unbinding of TPX2 could be ignored in our model.